# Learning diffractive optical communication around arbitrary opaque occlusions

Md Sadman Sakib Rahman[1,2,3], Tianyi Gan [1,3], Emir Arda Deger [1], Çağatay Işıl [1,2,3], Mona Jarrahi [1,3] & Aydogan Ozcan [1,2,3] ✉

Free-space optical communication becomes challenging when an occlusion blocks the light path. Here, we demonstrate a direct communication scheme, passing optical information around a fully opaque, arbitrarily shaped occlusion that partially or entirely occludes the transmitter's field-of-view. In this scheme, an electronic neural network encoder and a passive, all-optical diffractive network-based decoder are jointly trained using deep learning to transfer the optical information of interest around the opaque occlusion of an arbitrary shape. Following its training, the encoder-decoder pair can communicate any arbitrary optical information around opaque occlusions, where the information decoding occurs at the speed of light propagation through passive light-matter interactions, with resilience against various unknown changes in the occlusion shape and size. We also validate this framework experimentally in the terahertz spectrum using a 3D-printed diffractive decoder. Scalable for operation in any wavelength regime, this scheme could be particularly useful in emerging high data-rate free-space communication systems.

Traditionally radio frequency (RF) and microwave have dominated the area of wireless communication. To meet the growing need for faster data transfer rates, RF systems employ increasingly complex coding, multiple antennas, and higher carrier frequencies[1]. For example, by utilizing higher frequency bands, 6th generation (6G) technology is predicted to provide 100 to 1000 times faster speed than 5th generation (5G) systems deployed for wireless communication[2]. With ever-increasing data rates, maintaining the performance of these schemes will become more challenging. One possible solution is to shift to shorter wavelengths, such as the ultraviolet (UV), visible or infrared (IR) regions of the electromagnetic spectrum, which provide much wider bandwidths compared to radio waves or microwaves[1,3–5]. However, free-space optical communication becomes challenging when opaque occlusions block the light path. Non-line-of-sight (NLOS) communication, which exploits diffusely reflected waves from a nearby scattering medium, has been used as a way around the occlusion problem[6–10]. However, the adaptability of these solutions to

emerging optical communication techniques for channel capacity expansion faces challenges since even weak turbulence can cause a significant loss of information[10]. Furthermore, the low power efficiency arising from the weak scattering or diffuse reflection is another limitation of NLOS communication. Other NLOS systems, e.g., for imaging around corners, also exist[11–24]; these approaches, however, involve relatively slow and power-consuming digital methods for image reconstruction. Alternative methods have been developed for image transmission through thick (but transmitting) occlusions, including e.g., holography[25–27], adaptive wavefront control[28–30], and others[31,32]. However, many of these techniques also involve digital reconstruction of the information, often requiring iterative algorithms. Moreover, most of these are applicable for multiple-scattering media, and do not address situations, where the light path is either partially or entirely obstructed by opaque occlusions with zero light transmittance.

Here we demonstrate an optical architecture for directly communicating optical information of interest around zero-transmittance

[1]Electrical and Computer Engineering Department, University of California, Los Angeles, CA 90095, USA. [2]Bioengineering Department, University of California, Los Angeles, CA 90095, USA. [3]California NanoSystems Institute (CNSI), University of California, Los Angeles, CA 90095, USA. ✉e-mail: ozcan@ucla.edu

occlusions using electronic encoding at the transmitter and all-optical diffractive decoding at the receiver. In our scheme, an electronic neural network, trained in unison with an all-optical diffractive deco-der, encodes the message of interest to effectively bypass the opaque occlusion and be decoded at the receiver by an all-optical decoder, using passive diffraction through thin structured layers. This all-optical decoding is performed on the encoded wavefront that carries the optical information or the message of interest, after its obstruction by an arbitrarily shaped opaque occlusion. The diffractive decoder pro-cesses the secondary waves scattered through the edges of the opaque occlusion using a passive, smart material comprised of successive spatially engineered surfaces[33], and performs the reconstruction of the hidden information at the speed of light propagation through a thin diffractive volume that axially spans <100 × λ, where λ is the wave-length of the illumination light.

We show that this combination of electronic encoding and all-optical decoding is capable of direct optical communication between the transmitter and the receiver even when the opaque occlusion body entirely blocks the transmitter's field-of-view (FOV). We also report an experimental demonstration of this scheme using a 3D-printed dif-fractive decoder that operates at the terahertz spectrum. Furthermore, we demonstrate that this scheme could be configured to be misalignment-resilient as well as highly power efficient, reaching dif-fraction efficiencies of >50% at its output. In the case of opaque occlusions that change their size over time, we also report that the encoder neural network could be retrained to successfully commu-nicate with an existing diffractive decoder, without changing its phy-sical structure that is already deployed. We also show that our encoder/decoder framework can be jointly trained to be resilient against unknown, random dynamic changes in the occlusion size and/ or shape, without the need to retrain the encoder or the decoder. This makes the presented concept highly dynamic and easy to adapt to external and uncontrolled/unknown changes that might happen between the transmitter and receiver apertures. This framework can be extended for operation at different parts of the electromagnetic spectrum, and would find applications in emerging high-data-rate free-space communication technologies, under scenarios where different undesired structures occlude the direct channel of communication between the transmitter and the receiver.

## Results

A schematic depicting the optical communication scheme around an opaque occlusion with zero light transmittance is shown in Fig. 1a. The message to be transmitted, e.g., the image of an object, is fed to an electronic/digital neural network, which outputs a phase-encoded optical representation of the message. This code is imparted onto the phase of a plane-wave illumination, which is transmitted toward the decoder through an aperture that is partially or entirely blocked by an opaque occlusion. The scattered waves from the edges of the opaque occlusion travel toward the receiver aperture as secondary waves, where a diffractive decoder all-optically decodes the received light to directly reproduce the message/object at its output FOV. This decod-ing operation is completed as the light propagates through the thin decoder layers. For this collaborative encoding-decoding scheme, the electronic encoder neural network and the diffractive decoder are jointly trained in a data-driven manner for effective optical commu-nication, bypassing the fully opaque occlusion positioned between the transmitter aperture and the receiver.

Figure 1b, c provide a deeper look into the encoder and the decoder architectures used in this work. As shown in Fig. 1b, the con-volutional neural network (CNN) encoder is composed of several convolution layers, followed by a dense layer representing the enco-ded output. This dense layer output is rearranged into a 2D-array corresponding to the spatial grid that maps the phase-encoded transmitter aperture. We assumed that both the desired messages

and the phase codes to be transmitted comprise 28 × 28 pixels unless otherwise stated. The architecture of the encoder remains the same across all the designs reported in this paper. The architecture of the diffractive decoder, which decodes the transmitted and obstructed phase-encoded waves, is shown in Fig. 1c. This figure shows a dif-fractive decoder comprising $L = 3$ spatially engineered surfaces/layers (i.e., $S_1$, $S_2$ and $S_3$); however, in this work, we also report results for designs comprising diffractive decoders with $L = 1$ and $L = 5$ layers, used for comparison. Together with the encoder CNN parameters, the spatial features of the diffractive surfaces of the all-optical decoder are optimized to decode the encoded and blocked/obscured wavefront. In this work, we consider phase-only diffractive features, i.e., only the phase values of the features at each diffractive surface are trainable (see the 'Methods' section for details). Figure 1 also compares the performance of the presented electronic encoding and diffractive decoding scheme to that of a lens-based camera. As shown in Fig. 1d, the lens images reveal significant loss of information caused by the opaque occlusion in a standard camera system, showcasing the scale of the problem that is addressed through our proposed approach.

For all the models reported in this work, the data-driven joint training of the electronic encoder CNN and the diffractive decoder was accomplished by minimizing a structural loss function defined between the object (ground-truth message) and the diffractive deco-der output, using 55,000 images of handwritten digits from the MNIST[34] training dataset, augmented by 55,000 additional custom-generated images (see the 'Methods' section as well as Supplementary Fig. S1 for details). All our results come from blind testing with objects/ messages never used during training.

To bring more insights into the occlusion width $w_o$, we define the critical width $w_c$ as the minimum width of the occlusion at which no direct ray can reach the receiver aperture from the transmitter aper-ture; see Supplementary Fig. S2. In addition to the widths of the transmitter ($w_t$) and the receiver ($w_l$) apertures, this critical occlusion width $w_c$ is also a function of the ratio of the distances of the trans-mitter and the receiver from the occlusion, i.e., $d_{to}$ and $d_{ol}$, respec-tively; it can be written as $w_c = w_t \left(1 + d_{to}/d_{ol}\right)^{-1} + w_l \left(1 + d_{ol}/d_{to}\right)^{-1}$ as detailed in Supplementary Fig. S2. For all our simulations, $w_t \approx 59.73\lambda$, $w_l \approx 106.67\lambda$ and $d_{to}/d_{ol} = 1/8$ were used, resulting in $w_c \approx 64.95\lambda$. In our analyses and figures, we report the occlusion width $w_o$ as a fraction of $w_c$, where in some cases $w_o > w_c$, i.e., no direct ray reaches the receiver aperture from the transmitter aperture.

### Numerical analysis of diffractive optical communication around opaque occlusions

First, we compare, for various levels of opaque occlusions, the per-formance of trained encoder-decoder pairs with different diffractive decoder architectures in terms of the number of diffractive surfaces employed. Specifically, for each of the occlusion width values, i.e., $w_o = 32.0\lambda \approx 0.5w_c$, $w_o = 53.3\lambda \approx 0.8w_c$ and $w_o = 74.7\lambda \approx 1.15w_c$, we designed three encoder-decoder pairs, with $L = 1$, $L = 3$, and $L = 5$ dif-fractive layers within the decoders, and compared the performance of these designs for new handwritten digits in Fig. 2a. This blind testing refers to internal generalization because even though these particular test objects were never used in training, they are from the same dataset. As shown in Fig. 2a, even $L = 1$ designs can faithfully decode the message for optical communication around these various levels of occlusions. Furthermore, as the number of layers in the decoder increases to $L = 3$ or $L = 5$, the quality of the output also gets better. While the performance of the $L = 1$ design deteriorates slightly as $w_o$ increases, the $L = 3$ and $L = 5$ designs do not show any appre-ciable degradation in qualitative performance for such bigger occlusions. Note that for an occlusion size of $w_o = 74.7\lambda \approx 1.15w_c$, none of the ballistic photons can reach the receiver aperture since the opaque occlusion completely blocks any direct light ray from the encoding transmitter aperture. Nonetheless, the scattering from

the occlusion edges suffices for the encoder-decoder pair to communicate faithfully.

The learned encoder phase representations of the objects by different designs of Fig. 2a look completely random to the human eye. To gain more insights into the generalization of these designs, we performed dimensionality reduction analysis on these encoded phase patterns representing the input objects[35]. For this analysis, we prepared a dataset of size $9 \times 10,000 = 90,000$ comprising the encoded

phase objects corresponding to previously unseen 10,000 MNIST test images, for each one of these 9 designs shown in Fig. 2a. Subsequently, we applied an unsupervised dimensionality reduction algorithm, t-distributed stochastic neighbor embedding (t-SNE)[36], to learn a 2D manifold of these encoded phase patterns for all the encoder/decoder designs. A scatterplot of the projections of these encoded phase patterns on the learned manifold is presented in Fig. 2b. The clustering of these projections into 9 subgroups corresponding to the 9 different

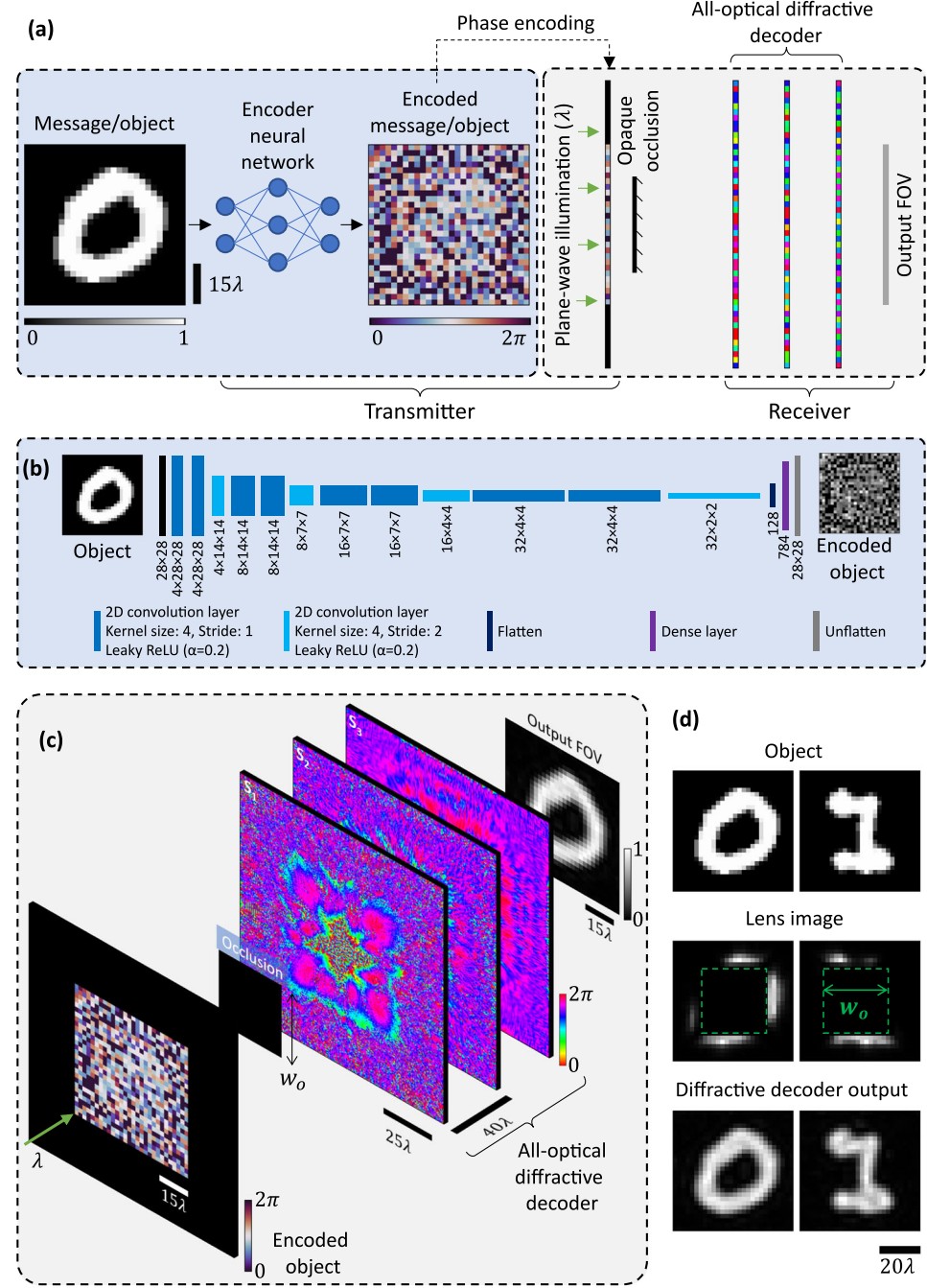

**Fig. 1 | Schematic of the optical communication framework around fully opaque occlusions using electronic encoding and diffractive all-optical decoding. a** An electronic neural network encoder and an all-optical diffractive decoder are trained jointly for communicating around an opaque occlusion. For a message/ object to be transmitted, the electronic encoder outputs a coded 2D phase pattern, which is imparted onto a plane wave at the transmitter aperture. The phase-encoded wave, after being obstructed and scattered by the fully opaque occlusion, travels to the receiver, where the diffractive decoder all-optically processes the

encoded information to reproduce the message on its output FOV. **b** The architecture used for the convolutional neural network (CNN) encoder throughout this work. **c** Visualization of different processes, such as the obstruction of the transmitted phase-encoded wave by the occlusion of width $w_o$ and the subsequent all-optical decoding performed by the diffractive decoder. The diffractive decoder comprises $L$ surfaces $(S_1, \cdots, S_L)$ with phase-only diffractive features. In this figure, $L = 3$ is illustrated as an example. **d** Comparison of the encoding-decoding scheme against conventional lens-based imaging.

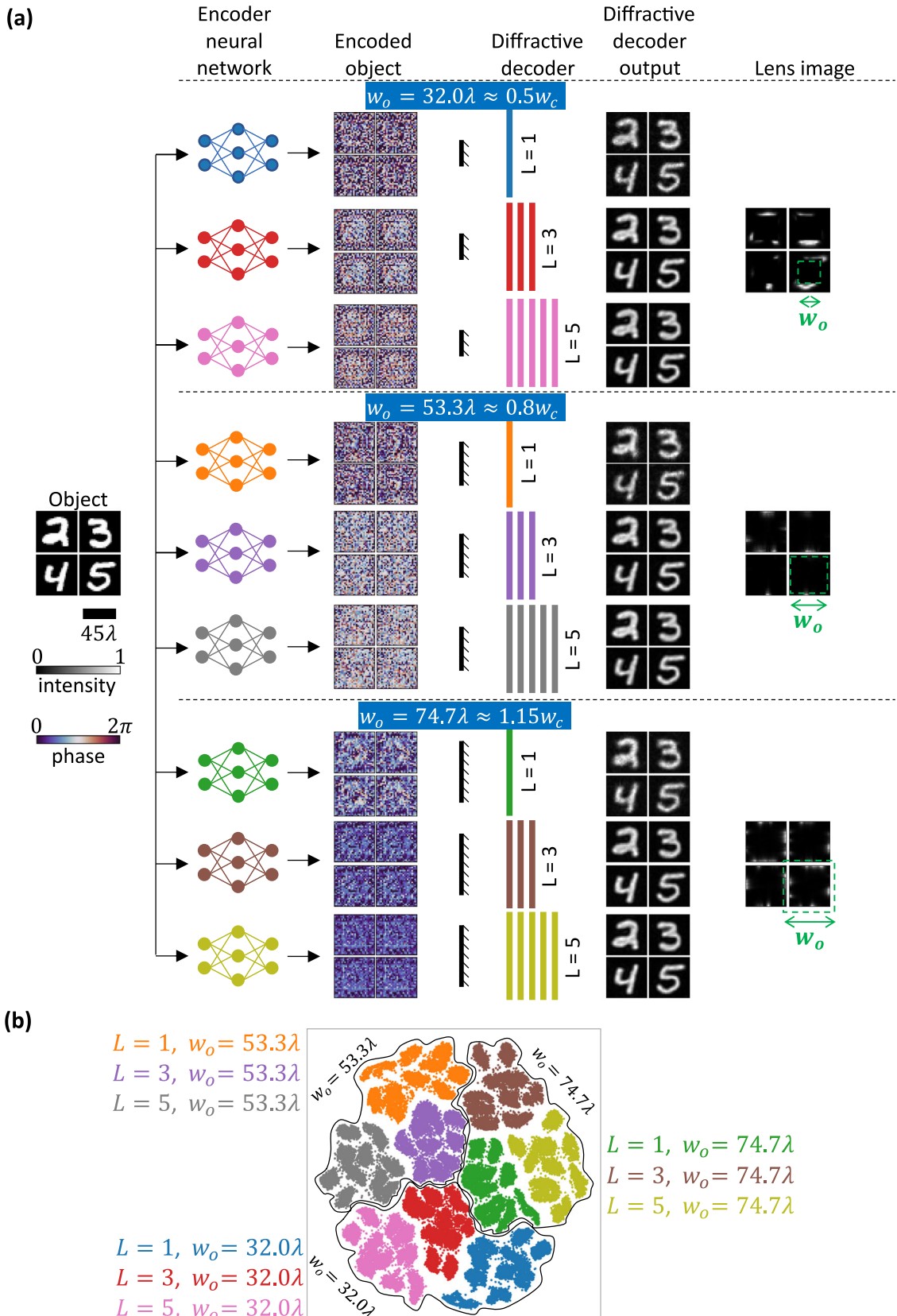

**Fig. 2 | Generalization of trained encoder-decoder pairs to previously unseen handwritten digit objects. a** For different values of the occlusion width $w_o$, the performances of trained encoder-decoder pairs with different numbers of decoder layers ($L$) are depicted for comparison. **b** t-SNE-based visualization of the electronic encoder outputs for the nine different designs of Fig. 2a.

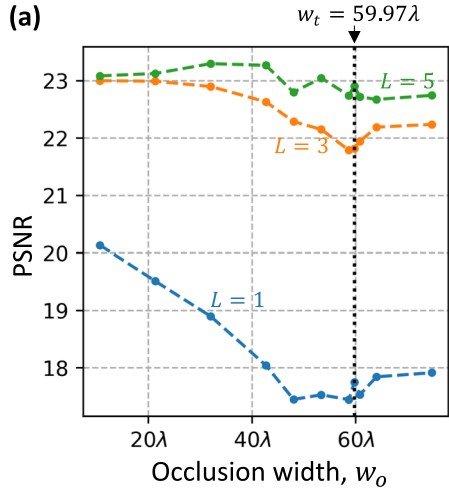

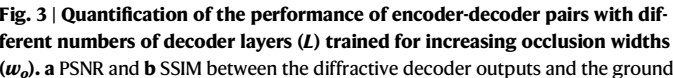

**Fig. 3 | Quantification of the performance of encoder-decoder pairs with different numbers of decoder layers ($L$) trained for increasing occlusion widths ($w_o$). a** PSNR and **b** SSIM between the diffractive decoder outputs and the ground-truth messages. The PSNR and SSIM values are calculated by averaging over 10,000 MNIST test images. $w_t$ refers to the width of the transmitter aperture.

designs with unique ($w_o$,$L$) attests to the generalization of these designs, indicating that the learned object representations in the phase space are specific to each architecture rather than being random. Figure 2b also shows the formation of three superclusters for each design corresponding to the three different occlusion sizes.

To supplement the results of Fig. 2, we also quantified the performance of different encoder-decoder pairs designed for increasing occlusion widths ($w_o$), in terms of peak signal-to-noise ratio (PSNR) and structural similarity index measure (SSIM)[37] averaged over 10,000 handwritten digits from the MNIST test set (never used before); see Fig. 3a, b, respectively. With increasing $w_o$, we see a larger decrease in the performance of $L=1$ designs compared to $L=3$ and $L=5$ designs. Interestingly, there is a slight improvement in the performance of $L=1$ and $L=3$ decoders as $w_o$ surpasses $w_t = 59.73\lambda$ (the transmitter aperture width); this improved level of performance is retained for $w_o>w_t$, the cause of which will be discussed later in our Discussion section.

Next, for the same designs reported in Fig. 2, we explored the external generalization of these encoder-decoder pairs by testing their performance on types of objects that were not represented in the training set; see Fig. 4. For this analysis, we randomly chose two images of fashion products from the Fashion-MNIST[38] test set (top) and two additional images from the CIFAR-10[39] test set (bottom). As shown in Fig. 4, our encoder-decoder designs show excellent generalization to these completely different object types. Although the decoder outputs of the $L=1$ decoder designs for $w_o = 53.3\lambda \approx 0.8w_c$ and $w_o = 74.7\lambda \approx 1.15w_c$ are slightly degraded, the objects are still recognizable at the output plane even for the complete blockage of the transmitter aperture by the occlusion.

We also investigated the ability of these designs to resolve closely separated features in their outputs. For this purpose, we transmitted test patterns consisting of four closely spaced dots, and the corresponding diffractive decoder outputs are shown in Fig. 5. For the top (bottom) pattern, the vertical/horizontal separation between the inner edges of the dots is 2.12$\lambda$ (4.24$\lambda$). None of the designs could resolve the dots separated by 2.12$\lambda$; however, the dots separated by 4.24$\lambda$ were resolved by all the encoder-decoder designs with good contrast, as can be seen from the cross-sections accompanying the output images in Fig. 5. It is to be noted that this resolution limit of 4.24$\lambda$ is due to the output pixel size, which was set as 2.12$\lambda$ in our simulations. The effective resolution of our encoder-decoder system can be further improved within the diffraction limit of light by using higher-resolution objects and a smaller pixel size during the training.

## Impact of phase bit depth on performance

Here, we study the effect of a finite bit-depth $b_q$ phase quantization of the encoder plane as well as the diffractive layers. For the results presented so far, we did not assume either to be quantized, i.e., an infinite bit-depth of phase quantization was assumed. For the $w_o = 32.0\lambda \approx 0.5w_c$, $L=3$ design (trained assuming an infinite bit-depth $b_{q,tr} = \infty$), the first row of Fig. 6a shows the impact of quantizing the encoded phase patterns as well as the diffractive layer phase values with a finite bit-depth $b_{q,te}$. This represents an attack on the design since the encoder CNN and the diffractive decoder were trained without such a phase bit-depth restriction; stated differently, they were trained with $b_{q,tr} = \infty$ and are now tested with finite levels of $b_{q,te}$. For the $b_{q,tr} = \infty$ designs, the output quality remains unaffected for $b_{q,te} = 8$; however, there is considerable degradation under $b_{q,te} = 4$, and we face complete failure with $b_{q,te} = 3$ and $b_{q,te} = 2$. However, this sharp performance degradation with decreasing $b_{q,te}$ can be amended by considering the finite bit-depth during training. To showcase this, we trained two additional designs with $w_o = 32.0\lambda$ and $L=3$ assuming finite bit-depths of $b_{q,tr} = 4$ and $b_{q,tr} = 3$; their blind testing performance with decreasing $b_{q,te}$ is reported in the second and third rows of Fig. 6a, respectively. Both of these designs show robustness against bit-depth reduction up to $b_{q,te} = 3$ (i.e., 8-level phase quantization at the encoder and decoder layers). However, even with $b_{q,te} = 2$ (only 4-level phase quantization), the outputs are still recognizable as shown in Fig. 6. We also quantified the performance (PSNR and SSIM) of these three designs ($b_{q,tr} = \infty$, $b_{q,tr} = 4$, $b_{q,tr} = 3$) for different $b_{q,te}$ levels; see Fig. 6b, c. These quantitative comparisons restate the same conclusion: training with a lower $b_{q,tr}$ results in robust encoder-decoder designs that preserve their optical communication quality despite a reduction in the bit-depth $b_{q,te}$, albeit with a relatively small sacrifice in the output performance.

## Impact of misalignments on performance

Next, we focus on the effect of physical misalignments on the performance of our framework for communication around opaque occlusions. First, we explore the effect of random misalignments of the physical layers of the diffractive decoder. For this analysis, we model the misalignments of the layers using random variables $\delta_{x,l} \sim Uniform(-\delta_{lat}, \delta_{lat})$, $\delta_{y,l} \sim Uniform(-\delta_{lat}, \delta_{lat})$ and $\delta_{z,l} \sim Uniform(-\delta_{ax}, \delta_{ax})$ where $\delta_{x,l}$, $\delta_{y,l}$ and $\delta_{z,l}$ denote the displacements of the diffractive layer $l$ from its nominal position along $x$, $y$ and $z$ directions, respectively; $l = 1, \cdots, L$. $\delta_{lat}$ and $\delta_{ax}$ are the parameters quantifying the degree of the lateral and axial random

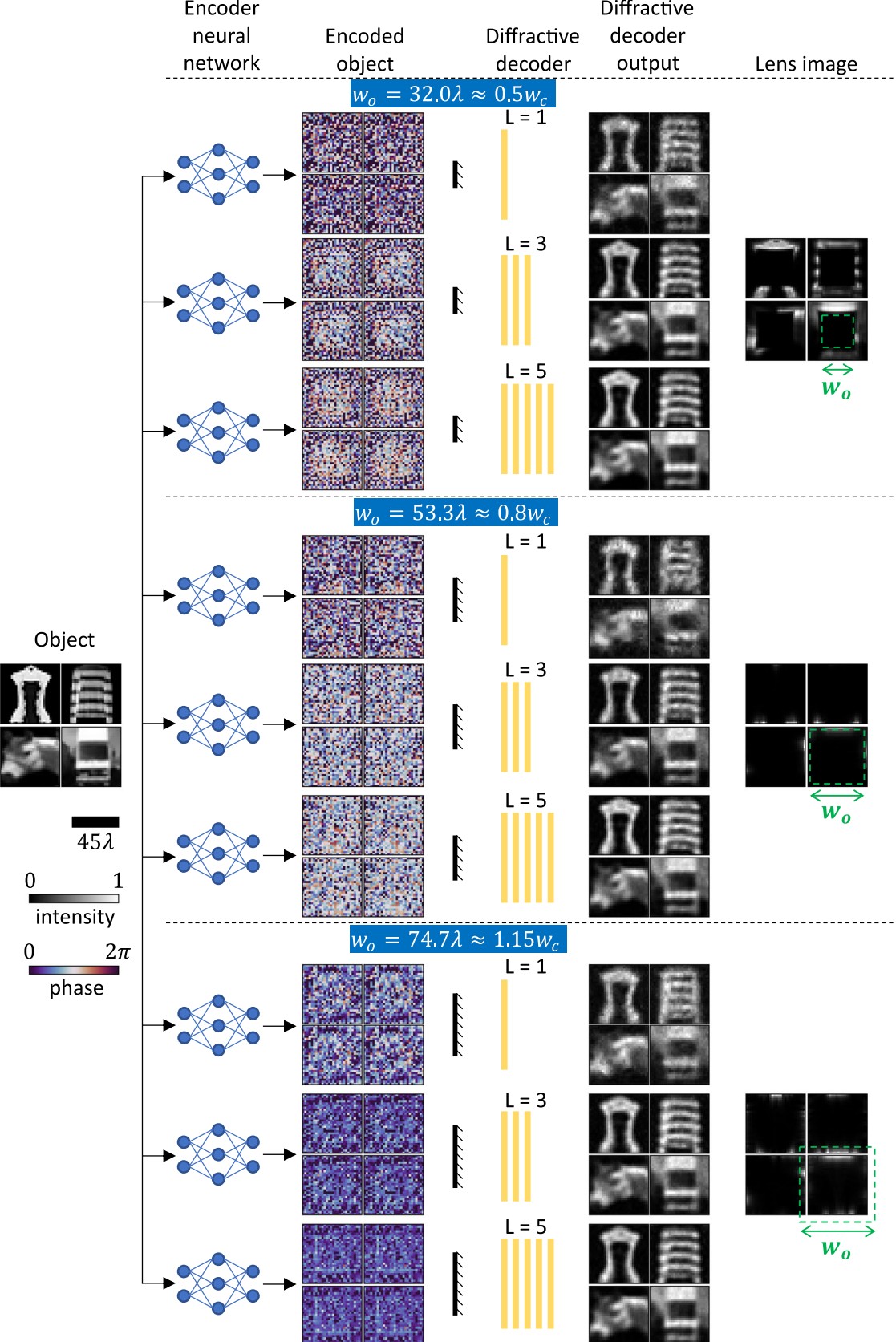

**Fig. 4 | Generalization of the trained encoder-decoder pairs to previously unseen objects.** Same as Fig. 2a, except that these results reflect external generalizations on object types different from those used during the training.

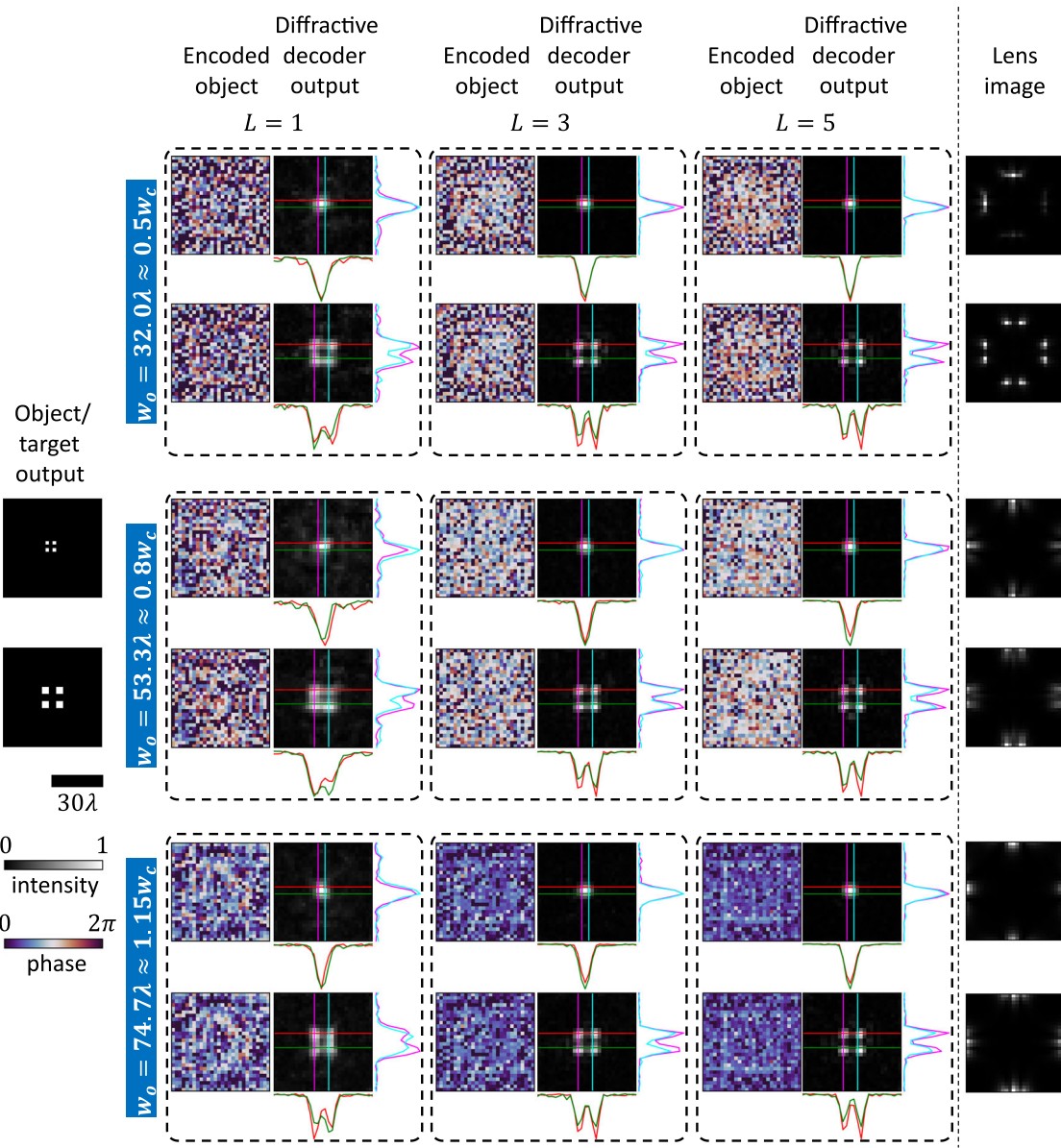

**Fig. 5 | Output resolution of diffractive decoders corresponding to L = 1, L = 3, and L = 5 designs trained for different occlusion widths ($w_o$).** As for the objects, the vertical/horizontal separation between the inner edges of the dots is 2.12λ for the test pattern on the top and 4.24λ for the one below. The diffractive decoder outputs are accompanied by cross-sections taken along the color-coded vertical/horizontal lines.

misalignments, respectively. In Supplementary Fig. S3, we present the effect of only the lateral random misalignments of the diffractive layers ($\delta_{lat} \geq 0$) assuming no axial misalignment ($\delta_{ax} = 0$). These results reveal that the design trained without taking such random lateral misalignments of the layers into consideration ($\delta_{lat,tr} = 0$) fails to successfully communicate through an opaque occlusion when tested with various levels of random lateral misalignments, i.e., $\delta_{lat,te} > 0$. This sensitivity to random physical misalignments can be improved by taking such misalignments into account during the design phase by training with $\delta_{lat,tr} > 0$. We can see from the same Supplementary Fig. S3 that the performance of the $\delta_{lat,tr} = 4\lambda$ design remains decent up to $\delta_{lat,te} = 8\lambda$, whereas for the $\delta_{lat,tr} = 8\lambda$ design, there is no perceptible degradation in the performance as $\delta_{lat,te}$ goes from 0 to 8λ. Supplementary Fig. S3b further reports, as a function of $\delta_{lat,te}$, the average PSNR and average SSIM values for these designs trained with different $\delta_{lat,tr}$, showing that the resilience of encoder-decoder designs against random lateral misalignments can be significantly improved by training with suitably chosen $\delta_{lat,tr}$, with a modest trade-off in communication performance.

The same conclusion also holds for axial random misalignments, as shown in Supplementary Fig. S4. It is to be noted that as the resilience to large random axial misalignments (e.g., $\delta_{ax,te} = 8\lambda$) is attained by training with $\delta_{ax,tr} > 0$, the decrease in performance for no misalignments ($\delta_{ax,te} = 0$) is virtually negligible, which is highly desired. Following a similar strategy, our jointly trained encoder-decoder pair designs can also be made resilient to lateral and axial random displacements of the opaque occlusion as illustrated in Supplementary Figs. S5 and S6.

## Output power efficiency

Next, we investigate the power efficiency of the optical communication scheme around opaque occlusions using jointly trained electronic encoder-diffractive decoder pairs. For this analysis, we defined the diffraction efficiency (DE) as the ratio of the optical power at the output FOV to the optical power departing the transmitter aperture. In Fig. 7a, we plot the diffraction efficiency of the same designs shown in Fig. 3, as a function of the occlusion size. These values are calculated by

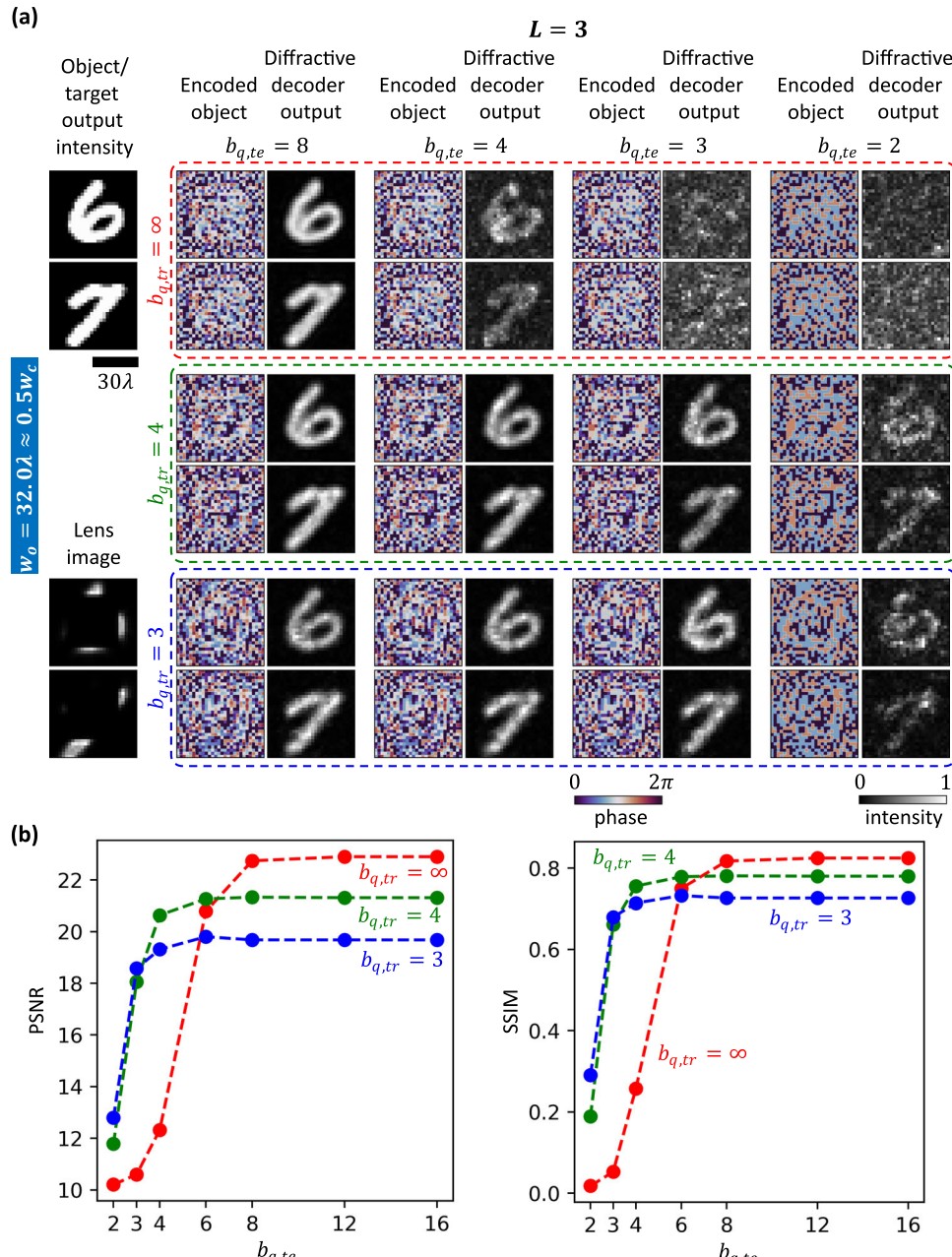

**Fig. 6 | Effect of the phase bit depth of the encoded object and the diffractive layer features on the performance of trained encoder-decoder pairs. a** Qualitative performance of the designs, which are trained assuming a certain phase quantization bit depth $b_{q,tr}$, reported as a function of the bit depth used during testing $b_{q,te}$. **b** For different $b_{q,tr}$, PSNR and SSIM values are plotted as a function of $b_{q,te}$. The PSNR and SSIM values are evaluated by averaging the results of 10,000 test images from the MNIST dataset.

averaging over 10,000 MNIST test images. These results reveal that the diffraction efficiency decreases monotonically with increasing occlusion width, as expected. Moreover, the diffraction efficiencies are relatively low, i.e., below or around 1%, even for small occlusions. However, this issue of low diffraction efficiency can be addressed in the design stage by adding to the training loss function an additional loss term that penalizes low diffraction efficiency (see the 'Methods' section, Eq. 5). Figure 7b depicts the improvement of diffraction efficiency resulting from increasing the weight ($\eta$) of this additive loss term during the training stage. For example, the $\eta = 0.02$ and $\eta = 0.1$ designs yield an average diffraction efficiency of 27.43% and 52.52%, respectively, while still being able to resolve various features of the target images as shown in Fig. 7c. This additive loss weight $\eta$ therefore provides a powerful mechanism for improving the output diffraction

efficiency significantly with a relatively small sacrifice in the image quality as exemplified in Fig. 7b, c.

**Occlusion shape**

So far, we have considered square-shaped opaque occlusions placed symmetrically around the optical axis. However, our proposed encoder-decoder approach is not limited to square-shaped occlusions and, in fact, can be used to communicate around any arbitrary occlusion shape. In Fig. 8, we show the performance comparison of four different trained encoder-decoder pairs for four different occlusion shapes, where the areas of the opaque occlusions were kept approximately the same. We can see that the shape of the occlusion does not have any perceptible effect on the output image quality. We also plot the average SSIM values calculated for these four models over 10,000

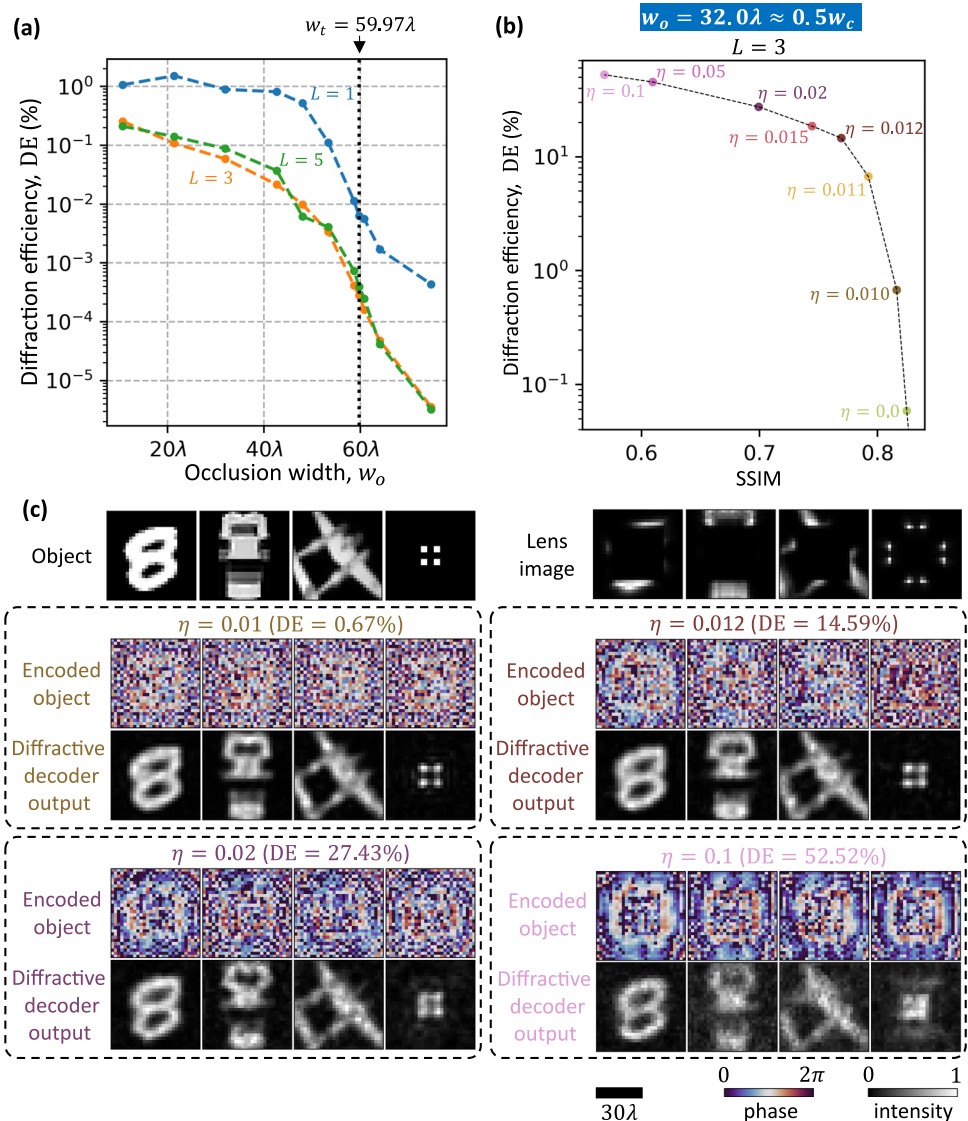

**Fig. 7 | Output power efficiency of the electronic encoding-diffractive decoding scheme for optical communication around fully opaque occlusions. a** Diffraction efficiency (DE) of the same designs shown in Fig. 3. **b** The trade-off between DE and SSIM achieved by varying the training hyperparameter $\eta$, i.e., the weight of an additive loss term used for penalizing low-efficiency designs. For these designs, $w_o = 32\lambda$ and $L = 3$ were used. The DE and SSIM values are calculated by averaging over 10,000 MNIST test images. **c** The performance of some of the designs shown in (b), trained with different $\eta$ values.

MNIST test images (internal generalization) as well as 10,000 Fashion-MNIST test images (external generalization) in Supplementary Fig. S7, which further confirm the success of our approach for different occlusion structures, including randomly shaped occlusions.

## Experimental validation

We experimentally validated the electronic encoding-diffractive decoding scheme for communication around opaque occlusion in the terahertz (THz) part of the spectrum ($\lambda = 0.75$ mm) using a 3D-printed single-layer ($L = 1$) diffractive decoder (see the 'Methods' section for details). We depict the setup used for this experimental validation in Fig. 9a. Figure 9b, c show the 3D printed components used to implement the encoded (phase) patterns, the opaque occlusion, and the diffractive decoder layer. Shown in Fig. 9c, the width of the transmitter aperture (dashed red square) housing the encoded phase patterns was selected as $w_t \approx 59.73\lambda$, whereas the width of the opaque occlusion (dashed green square) was $w_o \approx 32.0\lambda$ and the diffractive decoder layer (dashed blue square) width was selected as $w_l \approx 106.67\lambda$. The axial distances between the encoded object and the occlusion,

between the occlusion and the diffractive layer, and the diffractive layer and the output FOV were $\sim13.33\lambda$, $\sim106.67\lambda$, and $\sim40\lambda$, respectively. In Fig. 9d, we show the input objects/messages, the simulated lens images, and the simulated and experimental diffractive decoder output images for ten different handwritten digits randomly chosen from the test dataset. Our experimental results reveal that the CNN-based phase encoding followed by diffractive decoding resulted in successful communication of the intended objects/messages around the opaque occlusion (see the bottom row of Fig. 9d).

## Dynamic occlusions

So far, we have analyzed our framework for static occlusions that do not change over time. Here, we demonstrate the adaptability of our framework to situations where the occlusion shape/size can randomly change over time without our knowledge. In other words, we design encoder-decoder pairs which can communicate around opaque occlusions of varying unknown shapes, without any change to the encoder or the decoder. In Supplementary Video 1, we present our analysis depicting the performance of an $L = 3$ design as the shape of

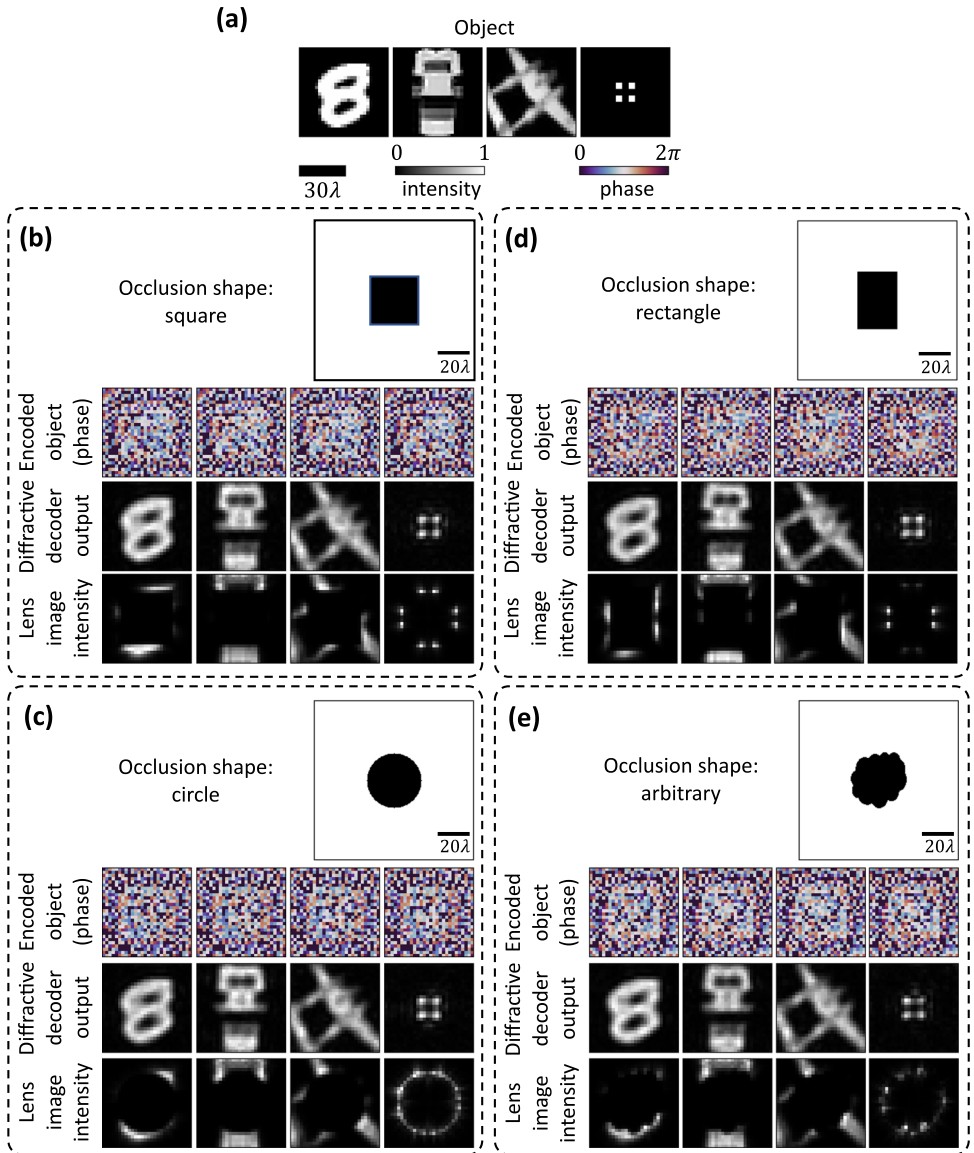

**Fig. 8 | Performance of encoder-decoder pairs trained for different opaque occlusion shapes.** The performances of four designs trained for different occlusion shapes, i.e., a square, a circle, a rectangle, and an arbitrary shape, are shown in panels (**b**)-(**e**) respectively while the objects are shown in **a**. The areas of these fully opaque occlusions are approximately equal.

the opaque occlusion randomly changes within the occlusion plane. For training and testing of this design, the occlusion shape was parameterized by $r_{max} \approx 17.6\lambda$, where the radii of the partial circles comprising the occlusions were randomly drawn from the distribution $Uniform(9r_{max}/11, r_{max})$; see the Methods section for details. As shown in Supplementary Video 1, the same electronic encoder and diffractive decoder successfully communicate the desired images of the objects even if the occlusion changes randomly. In Supplementary Videos 2 and 3, we show two additional $L = 3$ designs with $r_{max} \approx 29.3\lambda$ and $r_{max} \approx 41.1\lambda$, respectively, showcasing a decent performance despite a significant loss of information caused by the different opaque occlusions of varying random and unknown shapes.

We also experimentally demonstrated robust communication around opaque occlusions of varying, random shapes ($r_{max} \approx 18.1\lambda$) at $\lambda = 0.75$mm, using a fixed encoder-decoder pair with $L = 1$. The results of these experiments are presented in Fig. 10, where all the desired objects of interest were successfully communicated around two different occlusions by the same encoder-decoder design comprising a single-diffractive layer.

## Discussion

We modeled the scattering of light from the opaque occlusions with 2D cross-sections using the angular spectrum approach, covering a numerical aperture (NA) of 1.0 in air (see the Model subsection in the Methods). In general, any arbitrary fully opaque occlusion volume can be modeled, to a first-order approximation, as a string of scattering edges, forming a 3D loop of secondary waves. In our forward model, these 3D strings of scattering edges for any arbitrary opaque occlusion considered in our analysis were located at a common axial plane for ease of computing, and each point that makes up the scattering edge function communicated with the receiver aperture using traveling waves covering all the modes of free-space wave propagation (NA = 1). Using the Huygens–Fresnel principle, one can also extend the joint training of our encoder-decoder pairs to cover non-planar 3D strings of scattering edges representing any arbitrary occlusion volume; such cases, however, would take longer to numerically model and train using deep learning, especially if the axial coverage of the 3D string function that defines the scattering edges of a fully opaque volume is relatively large.

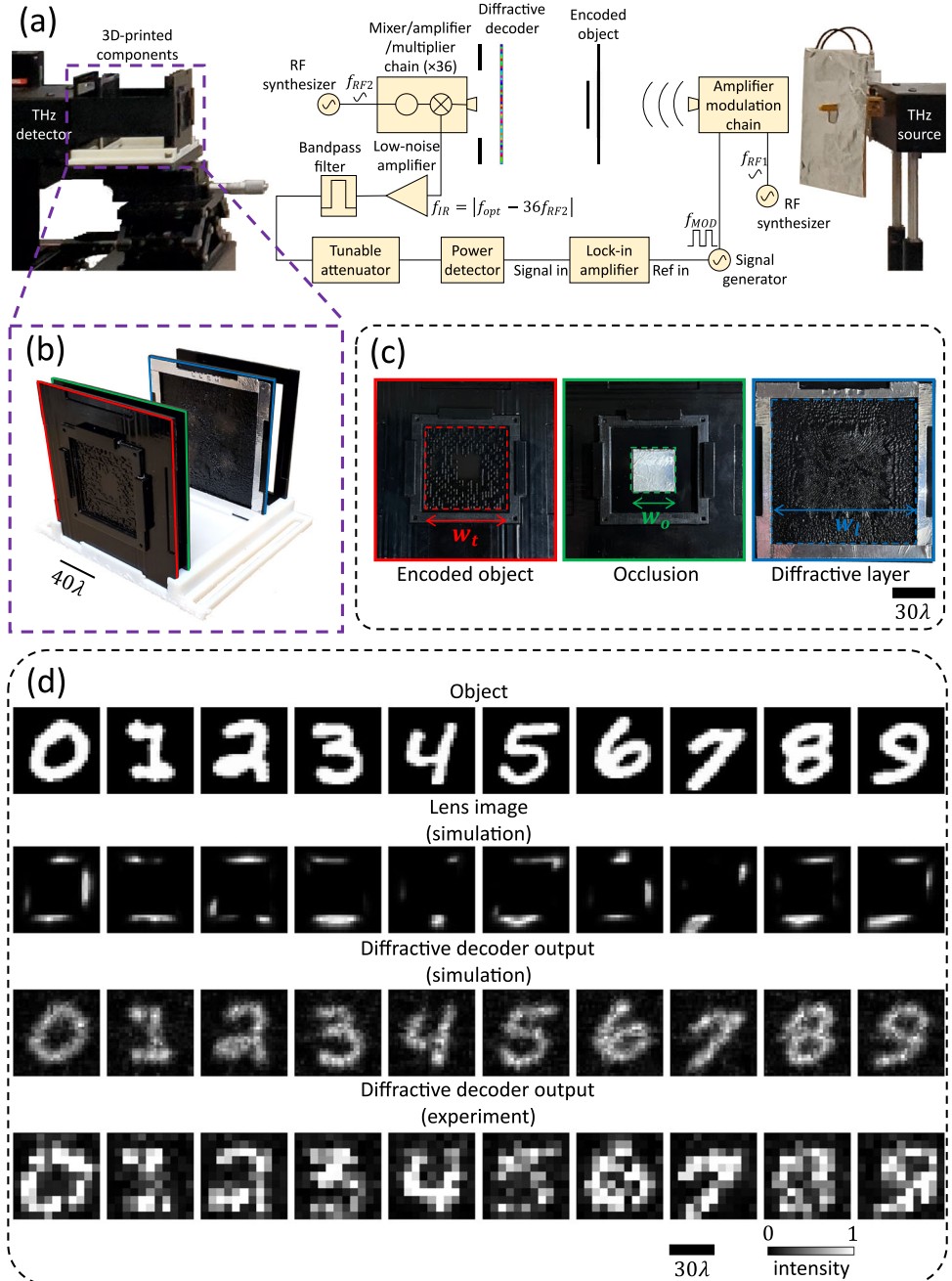

**Fig. 9 | Experimental results with an $L = 1$ design for an occlusion width of $w_o = 32\lambda \approx 0.5w_c$ operating at a wavelength of $\lambda = 0.75mm$. a** The terahertz setup comprising the source and the detector, together with the 3D-printed components used as the encoded phase objects, the occlusion, and the diffractive layer. **b** Assembly of the encoded phase objects, the occlusion, the diffractive layer, and the output aperture using a 3D-printed holder. **c** The encoded phase object (one example), the occlusion, and the diffractive layer are shown separately, housed inside the supporting frames. **d** Experimental diffractive decoder outputs (bottom row) for ten handwritten digit objects (top row), together with the corresponding simulated lens images (second row) and the diffractive decoder outputs (third row).

Our optical communication scheme using CNN-based encoding and diffractive all-optical decoding would be useful for the optical communication of information around opaque occlusions caused by existing or evolving structures. In case such occlusions grow in size as a function of time, the same diffractive decoder that is deployed as part of our communication link can still be used with only an update of the digital encoder CNN. To showcase this, in Supplementary Fig. S8, we illustrate an encoder-decoder design with $L = 3$ that was originally trained with an occlusion size of $w_o = 32.0\lambda$ (blue boxes), successfully communicating the input messages between the CNN-based phase transmitter aperture and the output FOV of the diffractive decoder

when the occlusion size remains the same, i.e., $w_o = 32.0\lambda$ (dashed blue box). The same figure also illustrates the failure of this encoder-decode pair once the size of the opaque occlusion grows to $w_o = 40.0\lambda$ (dotted blue box); this failure due to the (unexpectedly) increased occlusion size can be repaired without changing the deployed diffractive decoder layers by just retraining the CNN encoder part; see Supplementary Fig. S8, dashed green box.

The speed of optical communication through our encoder-decoder pair would be limited by the rate at which the encoded phase patterns (CNN outputs) can be refreshed or by the speed of the output detector-array, whichever is smaller. The transmission and the

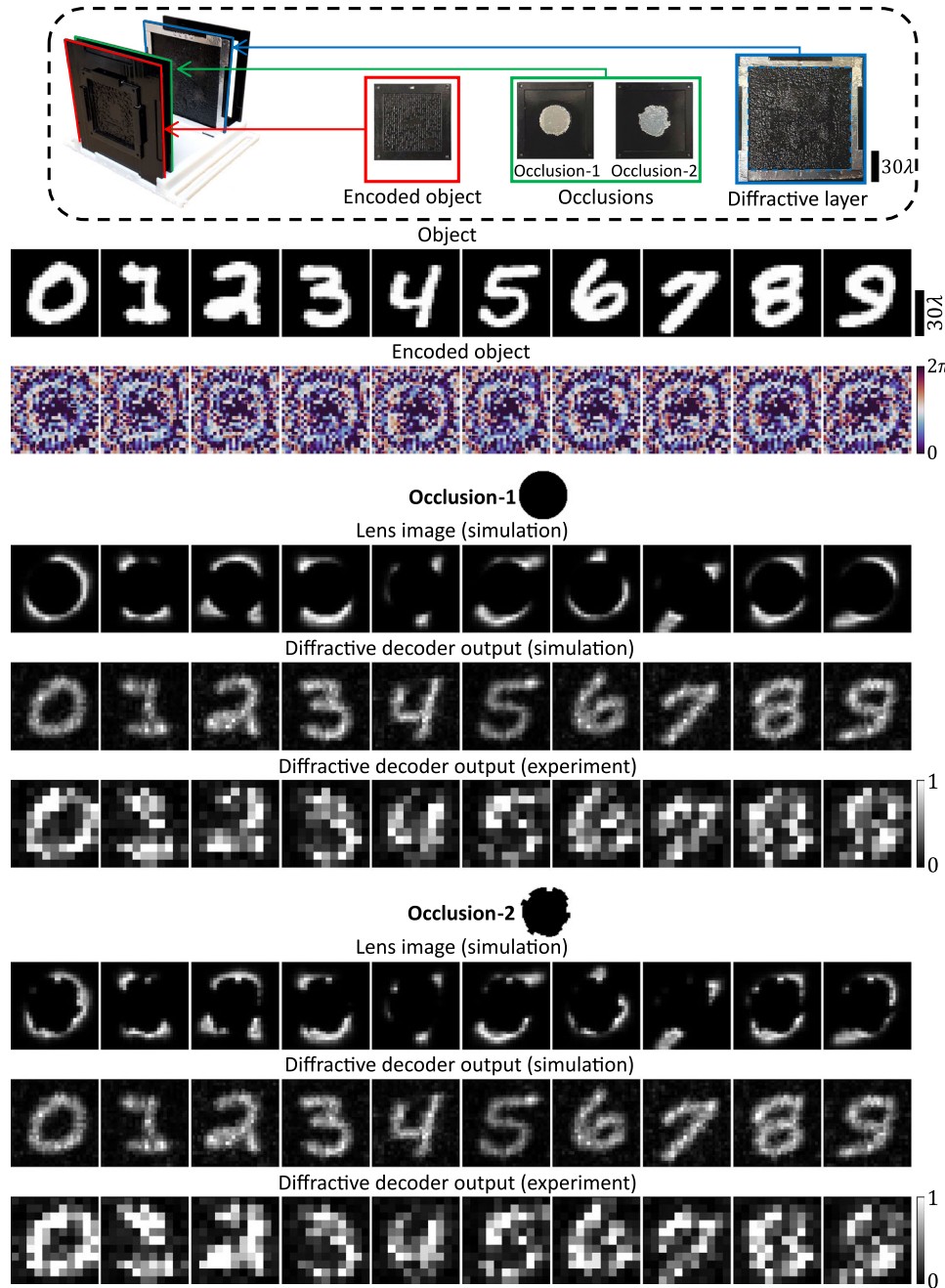

**Fig. 10 | Experimental results for communication around randomly changing opaque occlusions using the same electronic encoder and diffractive decoder;** $\lambda = 0.75$**mm.** The same electronically encoded objects and the 3D-printed single-layer diffractive decoder were used for communication around two different occlusions; $r_{max} = 18.1\lambda$.

decoding processes of the desired optical information/message occur at the speed of light propagation through thin diffractive layers and do not consume any external power (except for the illumination light). Therefore, the main power consuming steps in our architecture are the CNN inference, the transmitter of the encoded phase patterns and the detector-array operation.

The communication around occlusions using our scheme works even when the occlusion width is larger than the width of the transmitter aperture since it utilizes CNN-based phase encoding of information to effectively exploit the scattering from the edges of the occlusions. Surprisingly, as the occlusion width surpasses the transmitter aperture width ($w_t$), the performance of $L = 1$ and $L = 3$ designs slightly improved, as was seen in Fig. 3. This relative improvement might be explained by a switch in the mode of operation of our

encoder-decoder pair. When the opaque occlusions are smaller than the transmitter aperture, the pixels at the edges of the transmitter can communicate directly to the receiver aperture and therefore, they dominate the power balance. In this operation regime, as the occlusion size gets larger, the effective number of pixels at the transmitter aperture that directly communicates with the receiver/decoder gets smaller, causing a decline in the performance of the diffractive decoder. However, when the occlusion becomes larger than the transmitter aperture, none of the input pixels can dominate the power balance at the receiver end by communicating with it directly; instead, all the pixels of the encoder plane are forced to indirectly contribute to the receiver aperture through the edge scattering of the occlusion. This causes the performance to get better for occlusions larger than the transmitter aperture since effectively more pixels of the encoder plane

can contribute to the receiver aperture without a major power imbalance among these secondary wave-based contributions (through edge scattering). This turnaround in performance (i.e., the switching behavior between these two modes of operation) is not observed when the diffractive decoder has a deeper architecture (e.g., $L = 5$) since deeper decoders can effectively balance the ballistic photons that are transmitted from the edge pixels; consequently, edge-pixels of the transmitter aperture do not dominate the output signals even when they can directly see the receiver aperture since multiple layers of a deeper diffractive decoder act as a universal mode processor[40–43].

We would like to emphasize that the presented framework is also applicable for communication over larger distances ($d_{tl}$) between the transmitter and the receiver apertures. In Supplementary Fig. S9, we present results for communication over much larger axial distances of $d_{tl} = 600\lambda$ and $d_{tl} = 1200\lambda$ for two different occlusion widths. The diffractive decoder outputs reveal successful communication around the opaque occlusions for these larger distances; note, however, that as the axial distance gets even larger, the optical resolution of the decoder output will deteriorate because of the reduced NA of the communication system.

The success of the simpler decoder designs with $L = 1$ layer, as shown in Figs. 2–5 and 9, 10, begs the question of whether such an optical communication around opaque occlusions is also feasible with electronic encoding only, i.e., without diffractive decoding (see the 'Free space-only decoder' listed in Supplementary Fig. S10). To address this question, we trained two encoder-only designs, for $w_o = 32.0\lambda$ and $w_o = 53.3\lambda$, and compared their performance against $L = 1$ designs in Supplementary Fig. S11. The encoder-only architecture barely succeeds for $w_o = 32.0\lambda$ and fails drastically for $w_o = 53.3\lambda$, whereas $L = 1$ designs provide significantly better performance. In the same figure, we also evaluated two additional decoding approaches that do not use any trained diffractive decoders. In one of these approaches, we used a random (untrained) diffusive layer as the all-optical decoder, the phase profile of which was precisely known to the electronic encoder. We can see that the untrained diffusive layer fails to perform any meaningful decoding despite the presence of a trained electronic encoder. In the other approach that we used for comparison, we utilized a lens as the all-optical decoder, configured to perform the Fourier transform of the transmitter aperture. Similar to the 'Free space-only decoder', this Fourier lens-based decoder was also not as successful as our presented approach; see Supplementary Fig. S11. All these results and comparative analyses demonstrate the importance of complementing electronic encoding with diffractive decoding for effective communication around opaque occlusions.

We would also like to highlight some key differences between our approach and the transmission matrix-based approaches used to control light propagation through scattering media[44]. A transmission matrix-based approach relates the optical field at the receiver to the field at the transmitter in the presence of scattering, which can be measured or approximated. However, without the use of an optimized diffractive optical decoder architecture, the sole knowledge or accurate approximation of such a transfer matrix does not lend itself to the successful transfer of images or spatial information of interest around opaque occlusions with zero light transmittance; for example, Supplementary Fig. S11 illustrates that without an optimized diffractive decoder at the back-end, just an encoder optimization even with the precise knowledge of the transfer matrix of the system cannot perform successful image transfer around opaque occlusions. Moreover, our joint training approach for optimizing the electronic encoder−diffractive decoder pair is accomplished using the angular spectrum approach, which seamlessly blends all the propagating modes of optical information into our training. Using a wave propagation model instead of a known transmission matrix allows us to statistically incorporate various deviations in the forward model, which might randomly occur in real-world situations; this "vaccination" based training strategy builds design resilience against such random deviations in the physical system, as shown in Supplementary Figs. S3-S6 and Supplementary Videos 1-3. Measuring the transmission matrices corresponding to all the forward model states resulting from these random deviations would be impractical and, more importantly, would still not reveal competitive image transmission behavior around opaque occlusions without the use of a jointly trained diffractive optical decoder (see Supplementary Fig. S11). It should also be noted that the encoder neural network within our framework is not a surrogate for the transmission matrix. In fact, given the knowledge of how the optical field is scattered by the edges of an opaque occlusion located between the transmitter and the receiver, the electronic encoder learns an object representation model that successfully uses the edge scattering function of the occlusion to deliver the optical information to a jointly-trained diffractive decoder that all-optically converts this encoded and scattered information back to the desired object representation at the output, bypassing the zero transmission occlusion body.

Therefore, one of the important contributions of this work has been to establish an electronic-optical encoder-decoder communication pair that uses a string of scattering edges resulting from the topology of an opaque occlusion, forming a 3D loop of secondary waves. Several examples that we considered in this work solely used these strings of secondary waves in the form of edge scattering functions as the main source of optical information transmission; see, for example, Figs. 2a, 4 and 5 where $w_o > w_c$ and the opaque occlusion body entirely blocks the direct line-of-sight between the transmitter and the receiver apertures. Our framework can successfully transfer the target spatial information even in these cases, where the only communication channel between the transmitter and the receiver is the scattering from the occlusion edges.

Beyond optical image transmission around opaque occlusions, various other applications can potentially be enabled by the presented framework operating at different parts of the electromagnetic spectrum. For example, several mobile units/agents (such as autonomous robots) within a certain output region can be dynamically targeted even in the presence of occlusions that block the direct line-of-sight between the encoder/transmitter and the mobile receivers. In this scenario, the encoder can be dynamically updated to deliver optical radiation/power or information of interest to these mobile units/agents that are free to move within an output FOV. Another application of the presented concept could arise in the operation of wearable and implantable devices, which need to blend miniaturized optics and electronics for their operation. In such a scenario, our scheme could be useful, for example, to optically power/excite an array of implanted optical sensors by passing light around occlusions arising from the metal electronics or other opaque parts of the wearable/implantable system. Even more challenging applications could be envisioned using the presented framework to enable, for example, the visualization and detection of hidden objects sandwiched between two opaque occlusions, such as objects located between two walls or metal screens. In such applications, the first edge scattering function of the first opaque occlusion can be used by the encoder network to illuminate the hidden objects behind the first occlusion, whereas the second edge scattering function of the second opaque occlusion can be used to transmit the hidden objects' optical information to a jointly trained diffractive decoder network for all-optically revealing/reconstructing the information of the objects sandwiched between the opaque screens. This could have major implications for e.g., security and defense applications, enabling us to see/detect objects hidden between metal plates or partial walls. In such applications where the string of edge scattering function of each opaque occlusion is cascaded with the other edge scattering functions of successive opaque occlusions, the detection signal-to-noise ratio sets practical challenges, demanding high-power encoders/transmitters and high sensitivity output detectors.

## Methods

### Model

In our model, the message/object $m$ that is to be transmitted is fed to a CNN, which yields a phase-encoded representation $\psi$ of the message. The message is assumed to be in the form of an $N_{in} \times N_{in} = 28 \times 28$ pixel image. The coded phase $\psi$ is assumed to have dimension $N_{out} \times N_{out} = 28 \times 28$. The $N_{out} \times N_{out}$ phase elements are distributed over the transmitter aperture of area $w_t \times w_t$, where $w_t \approx 59.73\lambda$ and $\lambda$ is the illumination wavelength. The lateral width of each phase element/pixel is therefore $w_t/N_{out} \approx 2.12\lambda$. The phase-encoded input wave $\exp(j\psi)$ propagates a distance $d_{to} \approx 13.33\lambda$ to the plane of the opaque occlusion, where its amplitude is modulated by the occlusion function $o(x, y)$ such that:

$$o(x,y) = \begin{cases} 0, & |x| < \frac{w_o}{2}, |y| < \frac{w_o}{2} \\ 1, & \text{otherwise} \end{cases} \quad (1)$$

The encoded wave, after being obstructed and scattered by the occlusion, travels to the receiver through free space. At the receiver, the diffractive decoder all-optically processes and decodes the incoming wave to produce an all-optical reconstruction $\hat{m}'$ of the original message $m$ at its output FOV. We assume the receiver aperture, which coincides with the first layer of the diffractive decoder, to be located at an axial distance of $d_{ol} \approx 106.67\lambda$ away from the plane of the occlusion. The effective size of the independent diffractive features of each transmissive layer is assumed to be $0.53\lambda \times 0.53\lambda$, and each of the $L$ layers comprises $200 \times 200$ such diffractive features, resulting in a lateral width of $w_l \approx 106.67\lambda$ for the diffractive layers. The layer-to-layer separation is assumed to be $d_{ll} = 40\lambda$. The output FOV of the diffractive decoder is assumed to be $40\lambda$ away from the last diffractive layer and extend over an area $w_d \times w_d$, where $w_d \approx 59.73\lambda$.

The diffractive decoding at the receiver involves consecutive modulation of the received wave by the $L$ diffractive layers, each followed by propagation through the free space. The modulation of the incident optical wave on a diffractive layer is assumed to be realized passively by its height variations. The complex transmittance $\tilde{t}(x,y)$ of a passive diffractive layer is related to its height $h(x, y)$ according to:

$$\tilde{t} = \exp\left(j\frac{2\pi}{\lambda}(n+jk-1)h\right) = \exp\left(-\frac{2\pi k}{\lambda}h\right)\exp\left(j\frac{2\pi}{\lambda}(n-1)h\right) = a\exp(j\varphi) \quad (2)$$

where $n$ and $k$ are the refractive index and the extinction coefficient, respectively, of the diffractive layer material at $\lambda$; $a = \exp\left(-\frac{2\pi k}{\lambda}h\right)$ and $\varphi = \frac{2\pi}{\lambda}(n-1)h$ are the amplitude and the phase of the complex field transmittance, respectively. For our numerical simulations, we assume the diffractive layers to be lossless, i.e., $k = 0$, $a = 1$, unless stated otherwise.

The propagation of the optical fields through free space is modeled using the angular spectrum method[33,45], according to which the transformation of an optical field $u(x,y)$ after propagation by an axial distance $d$ can be computed as follows:

$$u(x,y;z=z_0+d) = \mathcal{F}^{-1}\left\{\mathcal{F}\{u(x,y;z=z_0)\} \times H(f_x, f_y; d)\right\} \quad (3)$$

where $\mathcal{F}$ ($\mathcal{F}^{-1}$) is the two-dimensional Fourier (Inverse Fourier) transform operator and $H(f_x, f_y; d)$ is the free-space transfer function for propagation by an axial distance $d$ defined as follows:

$$H(f_x, f_y; d) = \begin{cases} \exp\left(j\frac{2\pi}{\lambda}d\sqrt{1-(\lambda f_x)^2-(\lambda f_y)^2}\right), & f_x^2+f_y^2 < 1/\lambda^2 \\ 0, & \text{otherwise} \end{cases} \quad (4)$$

In our numerical analyses, the optical fields were sampled at an interval of $\delta \approx 0.53\lambda$ along both $x$ and $y$ directions and the Fourier (Inverse Fourier) transforms were implemented using the Fast Fourier Transform (FFT) algorithm.

For the lens-based imaging simulations reported in this work, the plane wave illumination was assumed to be amplitude modulated by the object placed at the transmitter aperture, and the (thin) lens is assumed to be placed at the same plane as the plane of the first diffractive layer in the encoding-decoding scheme, with the diameter of the lens aperture equal to the width of the diffractive layer, i.e., $w_l \approx 106.67\lambda$.

### Training

The diffractive decoder features were parameterized using the latent variables $h_{\text{latent}}$ such that the feature heights $h$ are related to $h_{\text{latent}}$ according to $h = h_{\max} \times \frac{1+\sin(h_{\text{latent}})}{2}$, where $h_{\max}$ is a hyperparameter denoting the maximum height variation. We used $h_{\max} = \frac{\lambda}{n-1}$ so that the corresponding maximum phase modulation was $\varphi_{\max} = 2\pi$.

The parameters of the encoder CNN and the diffractive decoder phase features were optimized by minimizing the loss function:

$$\mathcal{L} = \mathcal{L}_{\text{pixel}} + \eta\mathcal{L}_{\text{DE}} \quad (5)$$

where $\mathcal{L}_{\text{pixel}}$ is the mean squared error (MSE) between the pixels of the desired message $m$ and the pixels of the (scaled) decoded optical intensity $\hat{m} = \sigma\hat{m}'$, i.e.,

$$\mathcal{L}_{\text{pixel}} = \frac{1}{N_{in} \times N_{in}} \sum_{j=1}^{N_{in}} \sum_{i=1}^{N_{in}} \left(m_{ij} - \hat{m}_{ij}\right)^2 \quad (6)$$

The scaling factor $\sigma$ is defined as:

$$\sigma = \frac{\sum_{j=1}^{N_{in}} \sum_{i=1}^{N_{in}} m_{ij}\hat{m}'_{ij}}{\sum_{j=1}^{N_{in}} \sum_{i=1}^{N_{in}} \left(\hat{m}'_{ij}\right)^2} \quad (7)$$

The additive loss term $\mathcal{L}_{\text{DE}} = 1 - \text{DE}$, scaled by the weight $\eta$, is used to penalize against low diffraction efficiency models. DE is the diffraction efficiency, calculated as:

$$\text{DE} = \frac{1}{N_{in} \times N_{in}} \sum_{j=1}^{N_{in}} \sum_{i=1}^{N_{in}} \hat{m}'_{ij} \quad (8)$$

The training data comprised 110,000 examples: 55,000 images from the MNIST training set and 55,000 custom-prepared images; see Supplementary Fig. S1 for examples. The remaining 5000 images of the 60,000 MNIST training images, together with 5000 additional custom-prepared images, i.e., a total of 10,000 images, were used for validation. After the completion of each epoch, the average loss over the validation images was computed, and the model state corresponding to the smallest validation loss was selected as the ultimate design.

The electronic encoder-diffractive decoder models were implemented in TensorFlow[46] version 2.4 using the Python programming language and trained on a machine with Intel(R) Core(TM) i7-8700 CPU @ 3.20 GHz and NVIDIA GeForce GTX 1080 Ti GPU. The loss function was minimized using the Adam[47] optimizer for 50 epochs with a batch size of 4. The learning rate was initially 1e-3 and it decreased by a factor of 0.99 every 10,000 optimization steps. For the other parameters of the Adam optimizer, the default TensorFlow settings were used. The training time varied with the model size; for example, training a model with an $L = 3$ diffractive decoder took ~8 h.

The native TensorFlow implementations of PSNR and SSIM were used for computing these image comparison metrics between the message $m$ and the scaled diffractive decoder output $\hat{m}$.

## Experimental design

In our experiments, the wavelength of operation was $\lambda = 0.75$ mm. We used a single-layer diffractive decoder, i.e., $L = 1$, with $N = 200^2$ independent features and the width of each feature was $\sim 0.53\lambda \approx 0.40$ mm, resulting in an $\sim 80$ mm $\times 80$ mm diffractive layer. The width of the transmitter aperture accommodating the encoded phase messages was $w_t \approx 59.73\lambda \approx 44.8$ mm, same as the width of the output FOV $w_d$. The occlusion width was $w_o \approx 32\lambda \approx 24$ mm. The distance from the transmitter aperture to the occlusion plane was $d_{to} \approx 13.33\lambda \approx 10$ mm, while the diffractive layer was $d_{ol} \approx 106.67\lambda \approx 80$ mm away from the occlusion plane. The output FOV was $40\lambda \approx 30$ mm away from the diffractive layer.

The diffractive layers and the phase-encoded messages (CNN outputs) were fabricated using a 3D printer (Objet30 Pro, Stratasys Ltd). Similar to the implementation of the diffractive layer phase, the phase-encoded messages were implemented by height variations according to $h_o = \psi \frac{\lambda}{2\pi(n-1)}$. The height variations were applied on top of a uniform base thickness of 0.2 mm, used for mechanical support. The occlusion was realized by pasting aluminum on a 3D-printed substrate (see Fig. 9). The measured complex refractive index $n + jk$ of the 3D-printing material at $\lambda = 0.75$ mm was $1.6518 + j0.0612$.

While training the experimental model, the weight $\eta$ of the diffraction efficiency-related loss term was set to be zero. To make the experimental design robust against misalignments, we incorporated random lateral and axial misalignments of the encoded objects, the occlusion and the diffractive layer into the optical forward model during its training[48]. The random misalignments were modeled using the uniformly distributed random variables $\delta_x \sim Uniform(-0.5\lambda, 0.5\lambda)$, $\delta_y \sim Uniform(-0.5\lambda, 0.5\lambda)$ and $\delta_z \sim Uniform(-2\lambda, 2\lambda)$ representing the displacements of the encoded objects, the occlusion and the diffractive layer along $x$, $y$ and $z$ directions, respectively, from their nominal positions.

## Terahertz experimental setup

A WR2.2 modular amplifier/multiplier chain (AMC) in conjunction with a compatible diagonal horn antenna from Virginia Diodes Inc. was used to generate a continuous-wave (CW) radiation at 0.4 THz, by multiplying a 10 dBm RF input signal at $f_{RF1} = 11.1111$ GHz 36 times. To resolve low-noise output data through lock-in detection, the AMC output was modulated at a rate of $f_{MOD} = 1$ kHz. The exit aperture of the horn antenna was positioned ~60 cm away from the input (encoded object) plane of the 3D-printed diffractive decoder for the incident THz wavefront to be approximately planar. A single-pixel Mixer/AMC, also from Virginia Diodes Inc., was used to detect the diffracted THz radiation at the output plane. To down-convert the detected signal to 1 GHz, a 10 dBm local oscillator signal at $f_{RF2} = 11.0833$ GHz was fed to the detector. The detector was placed on an X-Y positioning stage consisting of two linear motorized stages from Thorlabs NRT100, and the output FOV was scanned using a 0.5 mm × 0.1 mm detector with a scanning interval of 2 mm. The down-converted signal was amplified, using cascaded low-noise amplifiers from Mini-Circuits ZRL-1150-LN + , by 40 dB and passed through a 1 GHz (+/−10 MHz) bandpass filter (KL Electronics 3C40-1000/T10-O/O) to filter out the noise from unwanted frequency bands. The filtered signal was attenuated by a tunable attenuator (HP 8495B) for linear calibration and then detected by a low-noise power detector (Mini-Circuits ZX47-60). The output voltage signal was read out using a lock-in amplifier (Stanford Research SR830), where the $f_{MOD} = 1$ kHz modulation signal served as the reference signal. The lock-in amplifier readings were converted to a linear scale according to the calibration results. To enhance the signal-to-noise ratio (SNR), a 2 × 2 binning was applied to the THz measurements. We also digitally enhanced the contrast of the measurements

by saturating the top 1% and the bottom 1% of the pixel values using the built-in MATLAB function *imadjust* and mapping the resulting image to a dynamic range between 0 and 1.

## Dynamic occlusion modeling

The dynamic occlusions were modeled as the union of 24 concentric and disjoint partial circles extending equal angles ($360^\circ / 24 = 15^\circ$) at the center. The radii of these partial circles were randomly sampled independently from the distribution $Uniform(9r_{max}/11, r_{max})$, where the parameter $r_{max}$ is a measure of the level of the opaque occlusion. At each training iteration, the occlusion was randomly sampled from the distribution defined by $r_{max}$. For the experimental demonstration reported in Fig. 10, we used $r_{max} \approx 18.1\lambda$.

## Reporting summary

Further information on research design is available in the Nature Portfolio Reporting Summary linked to this article.

## Data availability

All the data and methods needed to evaluate the conclusions in this work are present in the main text and the Supplementary Information. Any other relevant data are available from the authors upon request.

## Code availability

The deep learning models reported in this work used standard libraries and scripts that are publicly available in TensorFlow.

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

## Acknowledgements

**Funding**: The Ozcan Research Group at UCLA acknowledges the support of the U.S. Department of Energy (DOE), Office of Basic Energy Sciences, Division of Materials Sciences and Engineering under Award # DE-SC0023088. The Jarrahi Group at UCLA acknowledges the support of the Harvey Engineering Research Prize from the Institution of Engineering and Technology.

## Author contributions

A.O. conceived and initiated the research. M.S.S.R. conducted the numerical simulations and analyses. E.A.D. and C.I. assisted in numerical simulations. M.S.S.R. and T.G. performed the experimental validation. M.S.S.R. and A.O. wrote the manuscript; all the authors contributed to manuscript editing. A.O. and M.J. supervised the research.

## Competing interests

A.O., C.I. and M.S.S.R. are co-inventors of a pending provisional patent application on the presented method. The remaining authors declare no competing interests.
