## [Peer Review File · Nature Communications]

Learning Diffractive Optical Communication Around Arbitrary Opaque OcclusionsREVIEWER COMMENTS

Reviewer #1 (Remarks to the Author):

This is intriguing work that describes a joint digital-physical network to enable optical information transmission around opaque occlusions. The digital neural network (NN) acts as the encoder that makes the optical propagation through physical occluders "easier". The diffracted information from the edge of the occluders are being processed in real-time by an optical NN. Overall, I found the work to be highly novel. I only have a few minor questions.

1. Please elaborate on the modeling of the scattering from the edge of occluder. In general, can the authors comment on how the nature of the light occluder interaction (e.g. strong, weak, volumetric, thin object) will impact the design of the proposed framework?
2. Based on the visual appearance of the encoded phase, they look random. Perhaps the authors can perform some analysis using dimensional reduction analysis framework, e.g. UMAP, T-SNE, to explain the generalization of the NN. Ref: <https://opg.optica.org/oe/fulltext.cfm?uri=oe-29-2-2244&id=446557>
3. Can the authors provide some insights on the ratio between the lateral size of the occluder and size of the optical NN.
4. Extending the study in Fig. 8, what happens if the occluder is lateral or axially displaced?

Reviewer #2 (Remarks to the Author):

The authors propose a communication setup whereby an image is projected from a sender to a receiver over a distance, and this projection is made robust under partial occlusion by employing a diffractive encoding at the emitter and a diffractive decoding at the receiver.

This setting is somewhat contrived from a communications perspective -- in particular the idea of using a spatially resolved representation of information is far from optimal from a communications perspective -- it is less robust and efficient in the use of hardware resources compared to temporal encoding schemes.

As a result the occluder size is limited to be around the size of the aperture of the emitter and the receiver, which is quite small in absolute terms for communications wavelengths.

Regarding the occluder size, the written claims in the manuscript also seem inconsistent with the actual figures and data provided: the manuscript claims that the method works for full occlusion -- which would indicate blocking of all direct line of sight paths. However, the text lists the aperture of the received as 106λ , while the maximum occluder size in the results is 75λ . This would indicate that the most extreme obstruction is not a full occlusion.

Smaller comments:

- generally the apertures for emitter and receiver should be made more clear for each of the experiments. The occluder size should also be stated as a percentage occlusion instead of just a scale relative to wavelength.
- it would seem that the axial position of the occluder matters for the observed diffraction pattern.

Is the trained system able to handle different occluder positions?

- finally, the statement about the need for diffractive decoding is somewhat misleading -- the experiments show that a decoder is necessary, but of course a computational decoder would work as well (if not better due to the ability to use non-linear operators).

Reviewer #3 (Remarks to the Author):

In this manuscript the authors take a signal encoded in a wavefront (i.e. an image), convert it into a carefully chosen, but random-looking, phase pattern, block part of it with an obstacle, and then scatter it again on a number of fixed phase plates. What they show is that if the initial phase pattern was chosen appropriately, the result after the phase plates is a reconstruction of the original image.

The way they choose the specific phase pattern to encode the image is by training a neural network, such that the information necessary is routed around the occlusion, and is recombined into the desired image by the phase plates.

On the positive side, the authors made a lot of checks about how much the neural network could extrapolate outside its training set, but they failed to address the elephant in the room: how robust is this approach to changes in the occlusion and/or the misalignment of the phase plates? The point is that this experiment is conceptually equivalent to a simpler one: have the initial image hit a scattering medium, then place an occlusion that blocks some of the scattered light, make the transmitted light scatter again, detect the light, and then use the knowledge of the transmission matrix of the whole system to reconstruct the original image. The first and the second scattering mix the degrees of freedom insuring that: 1. enough of the information about the image passes around the occlusion, and 2. an intensity-only measurement contains enough of the phase information to allow for a successful reconstruction. Knowledge of the transmission matrix and the training of a neural network are largely equivalent in this context, and also putting the neural network at the beginning instead of the end is equivalent to put it in the end (due to reciprocity). The well known problem of this approach is the lack of transferability, i.e. small changes in the system (e.g. the exact shape and position of the occlusion) will exponentially decrease the quality of the reconstruction. And retraining the neural network every time the occlusion moved by one wavelength is not an answer to this problem.

I am always reluctant to ask the authors to perform new measurements, as I know how taxing it can be, but in this case I feel that a core point has been overlooked, and I can not recommend publication until it has been addressed.

Reviewer #4 (Remarks to the Author):

The manuscript entitled "Learning Diffractive Optical Communication Around Arbitrary Opaque Occlusions" by the Ozcan group presents a machine learning approach to optically transfer information in free space from the encoder to the decoder, even in the presence of large opaque occlusions. This approach optimizes the encoder and designs the diffractive decoder using machine learning techniques. The work can also be seen as using CNN to optimize the wavefront of the propagating beam, achieving the desired shape on the other side of the diffractive structure despite the occlusions. The diffractive structure is also optimized to achieve the best shape. The main contributions of this work are as follows: (1) the computational model to generate data for training and testing, which can subsequently be used for actual experimental demonstrations, and (2) the ML approach to design the diffractive encoder and determine the encoding phase pattern. The work is interesting and worth publishing. However, I cannot justify publishing it in Nat. Comm. My comments on the manuscript are as follows.

1. The work is relatively straightforward with wavefront control (phase encoder) because one can easily control the wavefront to make light go around the opaque occlusions, then hit the encoder and diffract to the output plane. I am not really convinced that the diffraction at the edge of the occlusion contributes significantly to the signal (i.e., more than the light directly from the encoder to the decoder). The encoder is significantly larger than the occlusion, so there is a direct path from the encoder to the diffractive decoder (though no direct path, i.e., ballistic photons, from the encoder to the output area), which is sufficient to control the light and achieve the desired outcome.
2. In the optimization process described at the end of the manuscript, where the result is weighted more for power efficiency, the light simply goes around the occlusion. One can simply adjust the phase to control the wavefront and achieve this. However, when the occlusion is large enough, the quality may be reduced or even fail.
3. In many other works (such as imaging or transmitting information around corners), the occlusion is, in fact, completely opaque, contrary to the authors' claim that "these are applicable for multiple scattering media that are transmissive and do not address situations where the light path is either partially or entirely obstructed by opaque occlusions with zero light transmittance."
4. In other works on imaging or transmitting information around corners, all the possible communication channels are through diffusive reflection surfaces, which play the role of the diffractive decoder in this work. The problem is even more challenging in those works because the diffusive reflection surface is unknown. In this context, the novelty of this work lies in the design of the diffractive media to achieve better performance, this is understandable because of the different purpose in this manuscript. However, there is no direct comparison with simple scattering media or diffusive reflection surfaces. The demonstration with a lens and an amplitude object (without an encoder or optimization) at the encoder is primarily to showcase the occlusion size and shape.
5. Besides the comparison with scattering media or diffusive reflection surfaces, a comparison with a lens is also an important demonstration. In the experiment with the lens, the authors should consider an approach that is exactly similar, with the only difference being the placement of the lens at the position of the diffractive encoder (without the need for optimization in decoder design). The lens should be placed in a manner that does not create a direct imaging system from the encoder plane to the output plane. One simple approach could be to position the output plane as the Fourier plane of the lens.
6. The next question pertains to the significance of free-space communication within distances of a few tens to 100 times the wavelength. Is this of significant importance?
7. For the encoder in this manuscript, we already know the diffractive media, and if we also know the size, shape, and positions of the occlusions, we can directly calculate the phase pattern to set on the encoder. This is similar to many other works.
8. The effectiveness of the machine learning (ML) approach employed here is also a question. If we want to use generated data for training, the requirement of matching between the real world and the virtual model needs to meet high levels of accuracy, which might only be achievable in the laboratory, as demonstrated in the manuscript. Alternatively, direct feedback communication channels need to be established from the output to the encoder. However, given the amount of training/testing data required for each configuration and the dynamics of real-world experiments, this approach could undermine the purpose of free-space communication around arbitrary opaque occlusions.
9. The experiment can be easily conducted using visible light, a phase-only spatial light modulator (SLM), a meta-surface (diffractive encoder), and a camera. Then possibly shows much more significant results compared to scattering media or diffusive reflection surface or lens, if any.

We sincerely thank the referees for their reviews and the constructive feedback that we have received on our manuscript “**Learning Diffractive Optical Communication Around Arbitrary Opaque Occlusions**” submitted to *Nature Communications*.

As detailed below, we have revised our manuscript in response to the reviewers’ comments. The original referee comments are shown in black color, whereas for ease of communication, our answers are provided in blue. The changes that we have made in the manuscript text are highlighted in yellow.

Summary of our Revisions:

We have revised our manuscript according to the reviewers’ comments, which will be detailed in our specific responses listed below.

- We have included additional analyses on the effect of random misalignments on the performance of our framework, and have successfully demonstrated strategies to improve the performance in the presence of physical random misalignments and uncertainties.
- We have also demonstrated, both numerically and experimentally, designs that are robust to unknown random changes in the opaque occlusion.
- We have also added comparative analyses on alternative decoding approaches as well as additional analyses on the generalization of the designed networks. We have revised the manuscript accordingly and added new results and discussions in response to the reviewers’ comments.

As a quick summary, the following items have been revised and/or added, all highlighted in yellow in our manuscript files:

Revised sub-sections:

- **Changes to the main text:**
 - Abstract
 - Introduction
 - Results
 - Discussion
 - Methods

New sub-sections added:

- New sub-sections added to the **Results** section:
 - Impact of misalignments on performance
 - Dynamic occlusions
- New sub-sections added to the **Methods** section:
 - Dynamic occlusion modeling

Renumbered Figures:

Previous	New
Supplementary Fig. S2	Supplementary Fig. S7
Supplementary Fig. S3	Supplementary Fig. S8
Supplementary Fig. S4	Supplementary Fig. S11

Revised Figures (the figure # refers to the number after renumbering):

- **Fig. 2:** Generalization of trained encoder-decoder pairs to previously unseen handwritten digit objects.
- **Supplementary Fig. S7:** Average SSIM values of the diffractive decoder outputs for the four designs of Fig. 8 (main text), calculated over 10,000 test images from the MNIST dataset (internal generalization) and 10,000 test images from the Fashion-MNIST dataset (external generalization).
- **Supplementary Fig. S11:** The performances of a known random diffusive decoder, free space-only decoder, and Fourier lens-based decoder (see Supplementary Fig. S10) are compared against the performance of $L = 1$ layer diffractive decoder for two different sizes of opaque occlusion (w_o).

New Figures Added:

- **Fig. 10:** Experimental results for communication around randomly changing opaque occlusions using the same electronic encoder and diffractive decoder.
 - **Supplementary Fig. S2:** The critical occlusion width w_c .
 - **Supplementary Fig. S3:** Impact of lateral random misalignments of the layers of the diffractive decoder on the performance of our framework for communication around opaque occlusions.
 - **Supplementary Fig. S4:** Impact of axial random misalignments of the layers of the diffractive decoder on the performance of our framework for communication around opaque occlusions.
 - **Supplementary Fig. S5:** Impact of lateral random displacements of the opaque occlusion on the performance of our framework for communication around opaque occlusions.
 - **Supplementary Fig. S6:** Impact of axial random displacements of the opaque occlusion on the performance of our framework for communication around opaque occlusions.
 - **Supplementary Fig. S9:** Performance of the electronic encoder and diffractive decoder framework for communication around opaque occlusions for large values of the axial distance d_{tl} between the transmitter and the receiver apertures.
 - **Supplementary Fig. S10:** Depiction of alternative schemes to diffractive all-optical decoding for communication around opaque occlusions.
-

New Supplementary Videos Added:

- **Supplementary Video 1:** Performance of an L=3 design for communication around opaque occlusions as the shape of the opaque occlusion randomly changes within the occlusion plane, while the electronic encoder and the diffractive decoder remain the same. $r_{max} \approx 17.6\lambda$.
 - **Supplementary Video 2:** Performance of an L=3 design for communication around opaque occlusions as the shape of the opaque occlusion randomly changes within the occlusion plane, while the electronic encoder and the diffractive decoder remain the same. $r_{max} \approx 29.3\lambda$.
 - **Supplementary Video 3:** Performance of an L=3 design for communication around opaque occlusions as the shape of the opaque occlusion randomly changes within the occlusion plane, while the electronic encoder and the diffractive decoder remain the same. $r_{max} \approx 41.1\lambda$.
-

Reviewer #1 (Remarks to the Author):

This is intriguing work that describes a joint digital-physical network to enable optical information transmission around opaque occlusions. The digital neural network (NN) acts as the encoder that makes the optical propagation through physical occluders "easier". The diffracted information from the edge of the occluders are being processed in real-time by an optical NN. Overall, I found the work to be highly novel. I only have a few minor questions.

-- We sincerely thank the reviewer for the positive evaluation.

1. Please elaborate on the modeling of the scattering from the edge of occluder. In general, can the authors comment on how the nature of the light occluder interaction (e.g. strong, weak, volumetric, thin object) will impact the design of the proposed framework?

-- Following the referee's suggestion, we have added the following paragraph to the Discussion section to shed more light on these details:

"...We modeled the scattering of light from the opaque occlusions with 2D cross-sections using the angular spectrum approach, covering a numerical aperture (NA) of 1.0 in air (see the Model subsection in the Methods section). In general, any arbitrary fully opaque occlusion volume can be modeled, to a first-order approximation, as a string of scattering edges, forming a 3D loop of secondary waves. In our forward model, these 3D strings of scattering edges for any arbitrary opaque occlusion considered in our analysis were located at a common axial plane for ease of computing, and each point that makes up the scattering edge function communicated with the receiver aperture using traveling waves covering all the modes of free-space wave propagation (NA=1). Using the Huygens–Fresnel principle, one can also extend the joint training of our encoder-decoder pairs to cover non-planar 3D strings of scattering edges representing any arbitrary occlusion volume; such cases, however, would take longer to numerically model and train using deep learning, especially if the axial coverage of the 3D string function that defines the scattering edges of a fully opaque volume is relatively large..."

2. Based on the visual appearance of the encoded phase, they look random. Perhaps the authors can perform some analysis using dimensional reduction analysis framework, e.g. UMAP, T-SNE, to explain the generalization of the NN. Ref: <https://opg.optica.org/oe/fulltext.cfm?uri=oe-29-2-2244&id=446557>

-- We thank the reviewer for this excellent suggestion. Following the referee's points, we have added Fig. 2b and the following text in the 'Results' section accordingly:

"...The learned encoder phase representations of the objects by different designs of Fig. 2a look completely random to the human eye. To gain more insights into the generalization of these designs, we performed dimensionality reduction analysis on these encoded phase patterns representing the input objects³⁵. For this analysis, we prepared a dataset of size $9 \times 10,000 = 90,000$ comprising the encoded phase objects corresponding to previously unseen 10,000 MNIST test images, for each one of these 9 designs shown in Fig. 2a. Subsequently, we applied an unsupervised dimensionality reduction algorithm, t-distributed stochastic neighbor embedding (t-SNE)³⁶, to learn a 2D manifold of these encoded phase patterns for all encoder/decoder designs. A scatterplot of the projections of these encoded phase patterns on the learned manifold is presented in Fig. 2b. The clustering of these

projections into 9 subgroups corresponding to the 9 different designs with unique (w_o, L) attests to the generalization of these designs, indicating that the learned object representations in the phase space are specific to each architecture rather than being random. Figure 2b also shows the formation of three superclusters for each design corresponding to the three different occlusion sizes.”

3. Can the authors provide some insights on the ratio between the lateral size of the occluder and size of the optical NN.

-- We have added these details in the Model subsection in the Methods section:

“...The effective size of the independent diffractive features of each transmissive layer is assumed to be $0.53\lambda \times 0.53\lambda$, and each of the L layers comprises 200×200 such diffractive features, resulting in a lateral width of $w_l \approx 106.67\lambda$ for the diffractive layers ... ”

Therefore, for an occlusion size of $w_o = 74.7\lambda$, the ratio would be 0.7.

To address the referee’s points and make it clearer for our readers, we have introduced a new reference quantity for the occlusion size (also see the new **Supplementary Fig. S2**), which was added in our revised manuscript, the ‘Results’ section:

“...To bring more insights into the occlusion width w_o , we define the critical width w_c as the minimum width of the occlusion at which no direct ray can reach the receiver aperture from the transmitter aperture; see Supplementary Fig. S2. In addition to the widths of the transmitter (w_t) and the receiver (w_l) apertures, this critical occlusion width w_c is also a function of the ratio of the distances of the transmitter and the receiver from the occlusion, i.e., d_{to} and d_{ol} , respectively; it can be written as $w_c = w_t(1 + d_{to}/d_{ol})^{-1} + w_l(1 + d_{ol}/d_{to})^{-1}$ as detailed in **Supplementary Fig. S2**. For all our simulations, $w_t \approx 59.73\lambda$, $w_l \approx 106.67\lambda$ and $d_{to}/d_{ol} = 1/8$; so $w_c \approx 64.95\lambda$. In our analyses and figures, we report the occlusion width w_o as a fraction of w_c , where in some cases $w_o > w_c$, i.e., no direct ray reaches the receiver aperture from the transmitter aperture.”

4. Extending the study in Fig. 8, what happens if the occluder is lateral or axially displaced?

-- To address the reviewer’s query, we have performed additional analyses on the effect of lateral and axial random displacements of the occlusion as well as the diffractive layers. We have also outlined strategies to make the designs more robust to such random misalignments.

Summarizing these new analyses, we have included four new supplementary figures, i.e., **Supplementary Figs. 3-6**, and added a new subsection titled ‘Impact of misalignments on performance’ in the ‘Results’ section, which reads as follows:

“...Next, we focus on the effect of physical misalignments on the performance of our framework for communication around opaque occlusions. First, we explore the effect of random misalignments of the physical layers of the diffractive decoder. For this analysis, we model the misalignments of the layers using random variables $\delta_{x,l} \sim \text{Uniform}(-\delta_{lat}, \delta_{lat})$, $\delta_{y,l} \sim \text{Uniform}(-\delta_{lat}, \delta_{lat})$ and $\delta_{z,l} \sim \text{Uniform}(-\delta_{ax}, \delta_{ax})$ where $\delta_{x,l}$, $\delta_{y,l}$ and $\delta_{z,l}$ denote the displacements of the diffractive layer l from its nominal position along x , y and z directions, respectively; $l = 1, \dots, L$. δ_{lat} and δ_{ax} are the parameters quantifying the degree of the lateral and axial random misalignments, respectively. In **Supplementary Fig. S3**, we present the effect of only the lateral random misalignments of the

diffractive layers ($\delta_{lat} \geq 0$) assuming no axial misalignment ($\delta_{ax} = 0$). These results reveal that the design trained without taking such random lateral misalignments of the layers into consideration ($\delta_{lat,tr} = 0$) fails to successfully communicate through an opaque occlusion when tested with various levels of random lateral misalignments, i.e., $\delta_{lat,te} > 0$. This sensitivity to random physical misalignments can be improved by taking such misalignments into account during the design phase by training with $\delta_{lat,tr} > 0$. We can see from the same **Supplementary Fig. S3** that the performance of the $\delta_{lat,tr} = 4\lambda$ design remains decent up to $\delta_{lat,te} = 8\lambda$, whereas for the $\delta_{lat,tr} = 8\lambda$ design, there is no perceptible degradation in the performance as $\delta_{lat,te}$ goes from 0 to 8λ . **Supplementary Fig. S3b** further reports, as a function of $\delta_{lat,te}$, the average PSNR and average SSIM values for these designs trained with different $\delta_{lat,tr}$, showing that the resilience of encoder-decoder designs against random lateral misalignments can be significantly improved by training with suitably chosen $\delta_{lat,tr}$, with a modest trade-off in communication performance.

The same conclusion also holds for axial random misalignments, as shown in **Supplementary Fig. S4**. It is to be noted that as the resilience to large random axial misalignments (e.g., $\delta_{ax,te} = 8\lambda$) is attained by training with $\delta_{ax,tr} > 0$, the decrease in performance for no misalignments ($\delta_{ax,te} = 0$) is virtually negligible, which is highly desired. Following a similar strategy, our jointly trained encoder-decoder pair designs can also be made resilient to lateral and axial random displacements of the opaque occlusion as illustrated in **Supplementary Figs. S5 and S6**."

Reviewer #2 (Remarks to the Author):

The authors propose a communication setup whereby an image is projected from a sender to a receiver over a distance, and this projection is made robust under partial occlusion by employing a diffractive encoding at the emitter and a diffractive decoding at the receiver.

This setting is somewhat contrived from a communications perspective -- in particular the idea of using a spatially resolved representation of information is far from optimal from a communications perspective -- it is less robust and efficient in the use of hardware resources compared to temporal encoding schemes.

As a result the occluder size is limited to be around the size of the aperture of the emitter and the receiver, which is quite small in absolute terms for communications wavelengths.

-- We sincerely thank the reviewer for the constructive feedback and valuable comments. We would like to emphasize that our work concerns the transfer of spatial information of objects or images around opaque occlusions, bypassing the necessity of conversion back and forth between the spatial domain and temporal sequences.

Regarding the occluder size, the written claims in the manuscript also seem inconsistent with the actual figures and data provided: the manuscript claims that the method works for full occlusion -- which would indicate blocking of all direct line of sight paths. However, the text lists the aperture of the received as 106 lambda, while the maximum occluder size in the results is 75 lambda. This would indicate that the most extreme obstruction is not a full occlusion.

-- This is an important discussion that the referee raised. **In fact, the largest occlusion that we reported is indeed a full occlusion, even by the definition suggested by the reviewer.**

To address the referee's points and make it clearer for our readers, we have introduced a new reference quantity for the occlusion size (also see the new **Supplementary Fig. S2**), which was added in our revised manuscript, the 'Results' section:

*"To bring more insights into the occlusion width w_o , we define the **critical width w_c** as the minimum width of the occlusion at which no direct ray can reach the receiver aperture from the transmitter aperture; see Supplementary Fig. S2. In addition to the widths of the transmitter (w_t) and the receiver (w_l) apertures, this critical occlusion width w_c is also a function of the ratio of the distances of the transmitter and the receiver from the occlusion, i.e., d_{to} and d_{ol} , respectively; it can be written as $w_c = w_t(1 + d_{to}/d_{ol})^{-1} + w_l(1 + d_{ol}/d_{to})^{-1}$ as detailed in **Supplementary Fig. S2**. For all our simulations, $w_t \approx 59.73\lambda$, $w_l \approx 106.67\lambda$ and $d_{to}/d_{ol} = 1/8$; so $w_c \approx 64.95\lambda$. In our analyses and figures, we report the occlusion width w_o as a fraction of w_c , where in some cases $w_o > w_c$, i.e., no direct ray reaches the receiver aperture from the transmitter aperture."*

The largest reported occlusion is $74.7\lambda \approx 1.15w_c$, and hence satisfies the full occlusion criterion, such that no direct ray can reach the receiver aperture from the transmitter aperture; please also refer to the newly added **Supplementary Fig. S2**.

Smaller comments:

- generally the apertures for emitter and receiver should be made more clear for each of the experiments. The occluder size should also be stated as a percentage occlusion instead of just a scale relative to wavelength.

-- We thank the reviewer for this constructive and important feedback. In addition to the existing mention of the transmitter and receiver aperture sizes in the 'Model' subsection of the 'Methods' section, we have included the following text in the 'Results' section:

"...For all our simulations, $w_t \approx 59.73\lambda$, $w_l \approx 106.67\lambda$ and $d_{to}/d_{ol} = 1/8$; so $w_c \approx 64.95\lambda$. In our analyses and figures, we report the occlusion width w_o as a fraction of w_c , where in some cases $w_o > w_c$, i.e., no direct ray reaches the receiver aperture from the transmitter aperture."

Following the referee's suggestion, as mentioned in the quoted text above, we have revised the main text and the figures to cite the occlusion width w_o as a fraction of the critical width w_c .

- it would seem that the axial position of the occluder matters for the observed diffraction pattern. Is the trained system able to handle different occluder positions?

-- To address the reviewer's query, we have performed additional analyses on the effect of lateral and axial random displacements of the occlusion as well as the diffractive layers. We have also outlined strategies to make the design robust to such physical random misalignments. We have **included four new supplementary figures, Supplementary Figs. 3-6**, and added a new subsection 'Impact of misalignments on performance' in the 'Results' section of our revised manuscript, quoted below:

"...Next, we focus on the effect of physical misalignments on the performance of our framework for communication around opaque occlusions. First, we explore the effect of random misalignments of the physical layers of the diffractive decoder. For this analysis, we model the misalignments of the layers using random variables $\delta_{x,l} \sim \text{Uniform}(-\delta_{lat}, \delta_{lat})$, $\delta_{y,l} \sim \text{Uniform}(-\delta_{lat}, \delta_{lat})$ and

$\delta_{z,l} \sim \text{Uniform}(-\delta_{ax}, \delta_{ax})$ where $\delta_{x,l}$, $\delta_{y,l}$ and $\delta_{z,l}$ denote the displacements of the diffractive layer l from its nominal position along x , y and z directions, respectively; $l = 1, \dots, L$. δ_{lat} and δ_{ax} are the parameters quantifying the degree of the lateral and axial random misalignments, respectively. In **Supplementary Fig. S3**, we present the effect of only the lateral random misalignments of the diffractive layers ($\delta_{lat} \geq 0$) assuming no axial misalignment ($\delta_{ax} = 0$). These results reveal that the design trained without taking such random lateral misalignments of the layers into consideration ($\delta_{lat,tr} = 0$) fails to successfully communicate through an opaque occlusion when tested with various levels of random lateral misalignments, i.e., $\delta_{lat,te} > 0$. This sensitivity to random physical misalignments can be improved by taking such misalignments into account during the design phase by training with $\delta_{lat,tr} > 0$. We can see from the same **Supplementary Fig. S3** that the performance of the $\delta_{lat,tr} = 4\lambda$ design remains decent up to $\delta_{lat,te} = 8\lambda$, whereas for the $\delta_{lat,tr} = 8\lambda$ design, there is no perceptible degradation in the performance as $\delta_{lat,te}$ goes from 0 to 8λ . **Supplementary Fig. S3b** further reports, as a function of $\delta_{lat,te}$, the average PSNR and average SSIM values for these designs trained with different $\delta_{lat,tr}$, showing that the resilience of encoder-decoder designs against random lateral misalignments can be significantly improved by training with suitably chosen $\delta_{lat,tr}$, with a modest trade-off in communication performance.

The same conclusion also holds for axial random misalignments, as shown in **Supplementary Fig. S4**. It is to be noted that as the resilience to large random axial misalignments (e.g., $\delta_{ax,te} = 8\lambda$) is attained by training with $\delta_{ax,tr} > 0$, the decrease in performance for no misalignments ($\delta_{ax,te} = 0$) is virtually negligible, which is highly desired. Following a similar strategy, our jointly trained encoder-decoder pair designs can also be made resilient to lateral and axial random displacements of the opaque occlusion as illustrated in **Supplementary Figs. S5 and S6**."

To conclude, the new Supplementary Figs. 5-6 demonstrate that our system can be trained to handle random and unknown changes in the opaque occlusion positions.

- finally, the statement about the need for diffractive decoding is somewhat misleading -- the experiments show that a decoder is necessary, but of course a computational decoder would work as well (if not better due to the ability to use non-linear operators).

-- We thank the reviewer for the opportunity to clarify this point. We would like to note that in our framework we employ **passive all-optical decoding** and entirely avoid relatively slow and power-consuming electronic decoders, not to mention the data communication requirement between the output imaging system and the digital processor. Hence, our statement is based upon comparison with other all-optical approaches that can be used for **passive decoding of spatial information**.

To further address these valuable comments, we explored two additional passive optical decoding alternatives to our presented diffractive all-optical decoders, both of which were confirmed to be significantly inferior in performance compared to our results:

- (i) 'Known random diffusive decoder' where the digital encoder is trained to collaborate with a random (i.e., untrained) diffusive layer whose phase profile is precisely known; and
- (ii) 'Fourier lens-based decoder' where the digital encoder is trained to collaborate with a lens configured to perform the Fourier transform of the transmitter aperture.

We have reported the results of our analyses in the revised **Supplementary Figs. S10 and S11**, and added the following text in the 'Discussion' section of our revised manuscript:

*"...In the same figure, we also evaluated two additional decoding approaches that do not use any trained diffractive decoders. In one of these approaches, we used a random (untrained) diffusive layer as the all-optical decoder, the phase profile of which was precisely "known" to the electronic encoder. We can see that the untrained diffusive layer fails to perform any meaningful decoding despite the presence of a trained electronic encoder. In the other approach that we used for comparison, we utilized a lens as the all-optical decoder, configured to perform the Fourier transform of the transmitter aperture. Similar to the 'Free space-only decoder', this Fourier lens-based decoder was also not as successful as our presented approach; see **Supplementary Fig. S11. All these results and comparative analyses demonstrate the importance of complementing electronic encoding with diffractive decoding for effective communication around opaque occlusions.**"*

These additional results are on top of our former analyses, which confirmed that a free-space only decoder with an electronic encoder network is also inferior compared to our results, as quoted from our manuscript:

*"...Finally, the success of the simpler decoder designs with $L = 1$ layer, as shown in Figs. 2-5 and 9-10, begs the question of whether such an optical communication around opaque occlusions is also feasible with electronic encoding only, i.e., without diffractive decoding (see the 'Free space-only decoder' listed in **Supplementary Fig. S10**). To address this question, we trained two encoder-only designs, for $w_o = 32.0\lambda$ and $w_o = 53.3\lambda$, and compared their performance against $L = 1$ designs in **Supplementary Fig. S11**. The encoder-only architecture barely succeeds for $w_o = 32.0\lambda$ and fails drastically for $w_o = 53.3\lambda$, whereas $L = 1$ designs provide significantly better performance."*

Reviewer #3 (Remarks to the Author):

In this manuscript the authors take a signal encoded in a wavefront (i.e. an image), convert it into a carefully chosen, but random-looking, phase pattern, block part of it with an obstacle, and then scatter it again on a number of fixed phase plates. What they show is that if the initial phase pattern was chosen appropriately, the result after the phase plates is a reconstruction of the original image.

The way they choose the specific phase pattern to encode the image is by training a neural network, such that the information necessary is routed around the occlusion, and is recombined into the desired image by the phase plates.

On the positive side, the authors made a lot of checks about how much the neural network could extrapolate outside its training set, but they failed to address the elephant in the room: how robust is this approach to changes in the occlusion and/or the misalignment of the phase plates?

-- We sincerely thank the reviewer for the positive and constructive remarks.

The point is that this experiment is conceptually equivalent to a simpler one: have the initial image hit a scattering medium, then place an occlusion that blocks some of the scattered light, make the transmitted light scatter again, detect the light, and then use the knowledge of the transmission matrix of the whole system to reconstruct the original image. The first and the second scattering mix the

degrees of freedom insuring that: 1. enough of the information about the image passes around the occlusion, and 2. a intensity-only measurement contains enough of the phase information to allow for a successful reconstruction. Knowledge of the transmission matrix and the training of a neural network are largely equivalent in this context, and also putting the neural network at the beginning instead of the end is equivalent to put it in the end (due to reciprocity). The well known problem of this approach is the lack of transferability, i.e. small changes in the system (e.g. the exact shape and position of the occlusion) will exponentially decrease the quality of the reconstruction. And retraining the neural network every time the occlusion moved by one wavelength is not an answer to this problem.

-- To address the concerns raised by the reviewer, we have reported a new design method for introducing robustness against random/unknown changes in the shape of the occlusion. We have also performed additional experiments to showcase the robustness of this approach to unknown changes in the occlusion. **New Supplementary Videos 1-3 and Fig. 10** have been added to showcase these results and analyses, together with a new subsection 'Dynamic occlusions' in the 'Results' section of our revised manuscript, quoted below:

*"...So far, we have analyzed our framework for static occlusions that do not change over time. Here, we demonstrate the adaptability of our framework to situations where the occlusion shape/size can randomly change over time without our knowledge. In other words, we design encoder-decoder pairs which can communicate around opaque occlusions of varying unknown shapes, without any change to the encoder or the decoder. In **Supplementary Video 1**, we present our analysis depicting the performance of an $L = 3$ design as the shape of the opaque occlusion randomly changes within the occlusion plane. For training and testing of this design, the occlusion shape was parameterized by $r_{max} \approx 17.6\lambda$, where the radii of the partial circles comprising the occlusions were randomly drawn from the distribution $Uniform(9r_{max}/11, r_{max})$; see the Methods section for details. As shown in **Supplementary Video 1**, the same electronic encoder and diffractive decoder successfully communicate the desired images of the objects even if the occlusion changes randomly. In **Supplementary Videos 2 and 3**, we show two additional $L = 3$ designs with $r_{max} \approx 29.3\lambda$ and $r_{max} \approx 41.1\lambda$, respectively, showcasing a decent performance despite a significant loss of information caused by the different opaque occlusions of varying random and unknown shapes.*

*We also experimentally demonstrated robust communication around opaque occlusions of varying, random shapes ($r_{max} \approx 18.1\lambda$) at $\lambda = 0.75mm$, using a fixed encoder-decoder pair with $L = 1$. The results of these experiments are presented in **Fig. 10**, where all the desired objects of interest were successfully communicated around two different occlusions by the same encoder-decoder design comprising a single-diffractive layer..."*

To further address the reviewer's query, we have performed additional analyses on the effect of lateral and axial random displacements of the occlusion as well as the diffractive layers. We have also outlined strategies to make the design robust to such physical random misalignments. We have **included four new supplementary figures, Supplementary Figs. 3-6**, and added a new subsection 'Impact of misalignments on performance' in the 'Results' section of our revised manuscript, quoted below:

"...Next, we focus on the effect of physical misalignments on the performance of our framework for communication around opaque occlusions. First, we explore the effect of random misalignments of

the physical layers of the diffractive decoder. For this analysis, we model the misalignments of the layers using random variables $\delta_{x,l} \sim \text{Uniform}(-\delta_{lat}, \delta_{lat})$, $\delta_{y,l} \sim \text{Uniform}(-\delta_{lat}, \delta_{lat})$ and $\delta_{z,l} \sim \text{Uniform}(-\delta_{ax}, \delta_{ax})$ where $\delta_{x,l}$, $\delta_{y,l}$ and $\delta_{z,l}$ denote the displacements of the diffractive layer l from its nominal position along x , y and z directions, respectively; $l = 1, \dots, L$. δ_{lat} and δ_{ax} are the parameters quantifying the degree of the lateral and axial random misalignments, respectively. In **Supplementary Fig. S3**, we present the effect of only the lateral random misalignments of the diffractive layers ($\delta_{lat} \geq 0$) assuming no axial misalignment ($\delta_{ax} = 0$). These results reveal that the design trained without taking such random lateral misalignments of the layers into consideration ($\delta_{lat,tr} = 0$) fails to successfully communicate through an opaque occlusion when tested with various levels of random lateral misalignments, i.e., $\delta_{lat,te} > 0$. This sensitivity to random physical misalignments can be improved by taking such misalignments into account during the design phase by training with $\delta_{lat,tr} > 0$. We can see from the same **Supplementary Fig. S3** that the performance of the $\delta_{lat,tr} = 4\lambda$ design remains decent up to $\delta_{lat,te} = 8\lambda$, whereas for the $\delta_{lat,tr} = 8\lambda$ design, there is no perceptible degradation in the performance as $\delta_{lat,te}$ goes from 0 to 8λ . **Supplementary Fig. S3b** further reports, as a function of $\delta_{lat,te}$, the average PSNR and average SSIM values for these designs trained with different $\delta_{lat,tr}$, showing that the resilience of encoder-decoder designs against random lateral misalignments can be significantly improved by training with suitably chosen $\delta_{lat,tr}$, with a modest trade-off in communication performance.

The same conclusion also holds for axial random misalignments, as shown in **Supplementary Fig. S4**. It is to be noted that as the resilience to large random axial misalignments (e.g., $\delta_{ax,te} = 8\lambda$) is attained by training with $\delta_{ax,tr} > 0$, the decrease in performance for no misalignments ($\delta_{ax,te} = 0$) is virtually negligible, which is highly desired. Following a similar strategy, our jointly trained encoder-decoder pair designs can also be made resilient to lateral and axial random displacements of the opaque occlusion as illustrated in **Supplementary Figs. S5 and S6**."

I am always reluctant to ask the authors to perform new measurements, as I know how taxing it can be, but in this case I feel that a core point has been overlooked, and I can not recommend publication until it has been addressed.

-- We sincerely thank the reviewer for giving us the opportunity to further strengthen our manuscript.

Reviewer #4 (Remarks to the Author):

The manuscript entitled "Learning Diffractive Optical Communication Around Arbitrary Opaque Occlusions" by the Ozcan group presents a machine learning approach to optically transfer information in free space from the encoder to the decoder, even in the presence of large opaque occlusions. This approach optimizes the encoder and designs the diffractive decoder using machine learning techniques. The work can also be seen as using CNN to optimize the wavefront of the propagating beam, achieving the desired shape on the other side of the diffractive structure despite the occlusions. The diffractive structure is also optimized to achieve the best shape. The main contributions of this work are as follows: (1) the computational model to generate data for training and testing, which can subsequently be used for actual experimental demonstrations, and (2) the ML approach to design the diffractive encoder and determine the encoding phase pattern.

The work is interesting and worth publishing. However, I cannot justify publishing it in Nat. Comm. My

comments on the manuscript are as follows.

1. The work is relatively straightforward with wavefront control (phase encoder) because one can easily control the wavefront to make light go around the opaque occlusions, then hit the encoder and diffract to the output plane. I am not really convinced that the diffraction at the edge of the occlusion contributes significantly to the signal (i.e., more than the light directly from the encoder to the decoder). The encoder is significantly larger than the occlusion, so there is a direct path from the encoder to the diffractive decoder (though no direct path, i.e., ballistic photons, from the encoder to the output area), which is sufficient to control the light and achieve the desired outcome.

-- We thank the reviewer for the constructive discussion. However, we would like to kindly disagree since our manuscript reported results for occlusion widths $w_o > w_c$ (e.g., $w_o = 1.15w_c$), where w_c is the minimum occlusion width at which no direct ray can reach the receiver (decoder) aperture from the transmitter (encoder) aperture. In other words, for these cases, there is no path for any unscattered ballistic photons to reach the decoder and the communication can only utilize scattered photons by the edges of the opaque occlusion.

To address the referee's points and make it clearer for our readers, we have introduced a new reference quantity for the occlusion size (also see the new **Supplementary Fig. S2**), which was added in our revised manuscript, the 'Results' section:

*"...To bring more insights into the occlusion width w_o , we define the **critical width w_c** as the minimum width of the occlusion at which no direct ray can reach the receiver aperture from the transmitter aperture; see **Supplementary Fig. S2**. In addition to the widths of the transmitter (w_t) and the receiver (w_l) apertures, this critical occlusion width w_c is also a function of the ratio of the distances of the transmitter and the receiver from the occlusion, i.e., d_{to} and d_{ol} , respectively; it can be written as $w_c = w_t(1 + d_{to}/d_{ol})^{-1} + w_l(1 + d_{ol}/d_{to})^{-1}$ as detailed in **Supplementary Fig. S2**. For all our simulations, $w_t \approx 59.73\lambda$, $w_l \approx 106.67\lambda$ and $d_{to}/d_{ol} = 1/8$; so $w_c \approx 64.95\lambda$. In our analyses and figures, we report the occlusion width w_o as a fraction of w_c , where in some cases $w_o > w_c$, i.e., no direct ray reaches the receiver aperture from the transmitter aperture."*

The largest reported occlusion in our manuscript is $74.7\lambda \approx 1.15w_c$, and hence satisfies the full occlusion criterion, such that no direct ray can reach the receiver aperture from the transmitter aperture; please also refer to the newly added **Supplementary Fig. S2.**

2. In the optimization process described at the end of the manuscript, where the result is weighted more for power efficiency, the light simply goes around the occlusion. One can simply adjust the phase to control the wavefront and achieve this. However, when the occlusion is large enough, the quality may be reduced or even fail.

-- We kindly disagree with the reviewer's comments/conclusions as they conflict with the results that we presented in the original submission: As we showed in **Supplementary Fig. S4 of our initial submission (revised figure # **Supplementary Fig. S11**)**, the encoder-only designs with no diffractive decoding perform much worse than our designs with digital encoders complemented by diffractive decoders. Please refer to the results corresponding to 'Free space-only decoder' in the revised **Supplementary Fig. S11**.

3. In many other works (such as imaging or transmitting information around corners), the occlusion is, in fact, completely opaque, contrary to the authors' claim that "these are applicable for multiple scattering media that are transmissive and do not address situations where the light path is either partially or entirely obstructed by opaque occlusions with zero light transmittance."

-- We thank the reviewer for these valuable comments. The initial introduction sentence that the referee points to was based on the observation that the works that we found in the literature on image transmission around opaque occlusions use the term 'opaque' to describe very thick samples where the photons undergo multiple scattering, not an obstacle with zero light transmittance (such as ours) 'killing' a significant fraction of the photons. Following the referee's criticism, we have revised the corresponding introduction sentence as such:

"...Moreover, most of these are applicable for multiple-scattering media, and do not address situations, where the light path is either partially or entirely obstructed by opaque occlusions with zero light transmittance."

4. In other works on imaging or transmitting information around corners, all the possible communication channels are through diffusive reflection surfaces, which play the role of the diffractive decoder in this work. The problem is even more challenging in those works because the diffusive reflection surface is unknown. In this context, the novelty of this work lies in the design of the diffractive media to achieve better performance, this is understandable because of the different purpose in this manuscript. However, there is no direct comparison with simple scattering media or diffusive reflection surfaces. The demonstration with a lens and an amplitude object (without an encoder or optimization) at the encoder is primarily to showcase the occlusion size and shape.

5. Besides the comparison with scattering media or diffusive reflection surfaces, a comparison with a lens is also an important demonstration. In the experiment with the lens, the authors should consider an approach that is exactly similar, with the only difference being the placement of the lens at the position of the diffractive encoder (without the need for optimization in decoder design). The lens should be placed in a manner that does not create a direct imaging system from the encoder phase to the output plane. One simple approach could be to position the output plane as the Fourier plane of the lens.

-- We address these comments 4 and 5 together since they are related.

To address these valuable comments, we explored two additional alternatives to our presented diffractive all-optical decoders, both of which were confirmed to be significantly inferior in performance compared to our results:

- (i) 'Known random diffusive decoder' where the digital encoder is trained to collaborate with a random (i.e., untrained) diffusive layer whose phase profile is precisely known; and
- (ii) 'Fourier lens-based decoder' where the digital encoder is trained to collaborate with a lens configured to perform the Fourier transform of the transmitter aperture.

We have reported the results of our analyses in the revised **Supplementary Figs. S10 and S11**, and added the following text in the 'Discussion' section of our revised manuscript:

“...In the same figure, we also evaluated two additional decoding approaches that do not use any trained diffractive decoders. In one of these approaches, we used a random (untrained) diffusive layer as the all-optical decoder, the phase profile of which was precisely “known” to the electronic encoder. We can see that the untrained diffusive layer fails to perform any meaningful decoding despite the presence of a trained electronic encoder. In the other approach that we used for comparison, we utilized a lens as the all-optical decoder, configured to perform the Fourier transform of the transmitter aperture. Similar to the ‘Free space-only decoder’, this Fourier lens-based decoder was also not as successful as our presented approach; see **Supplementary Fig. S11**. **All these results and comparative analyses demonstrate the importance of complementing electronic encoding with diffractive decoding for effective communication around opaque occlusions.**”

These additional results are on top of our former analyses, which confirmed that a free-space only decoder with an electronic encoder network is also inferior compared to our results, as quoted from our manuscript:

“...Finally, the success of the simpler decoder designs with $L = 1$ layer, as shown in Figs. 2-5 and 9-10, begs the question of whether such an optical communication around opaque occlusions is also feasible with electronic encoding only, i.e., without diffractive decoding (see the ‘Free space-only decoder’ listed in **Supplementary Fig. S10**). To address this question, we trained two encoder-only designs, for $w_o = 32.0\lambda$ and $w_o = 53.3\lambda$, and compared their performance against $L = 1$ designs in **Supplementary Fig. S11**. The encoder-only architecture barely succeeds for $w_o = 32.0\lambda$ and fails drastically for $w_o = 53.3\lambda$, whereas $L = 1$ designs provide significantly better performance.”

6. The next question pertains to the significance of free-space communication within distances of a few tens to 100 times the wavelength. Is this of significant importance?

-- We would like to emphasize that the applicability of our framework is not limited to communication within distances of a few tens to 100 times the wavelength. We have demonstrated this in our revised manuscript with new analyses and a new **Supplementary Fig. S9** by reporting designs for communication over axial distances of 600λ and 1200λ . We have also added the following paragraph in the ‘Discussion’ section:

“... We would like to emphasize that the presented framework is also applicable for communication over larger distances (d_{tl}) between the transmitter and the receiver apertures. In **Supplementary Fig. S9**, we present results for communication over much larger axial distances of $d_{tl} = 600\lambda$ and $d_{tl} = 1200\lambda$ for two different occlusion widths. The diffractive decoder outputs reveal successful communication around the opaque occlusions for these larger distances; note, however, that as the axial distance gets even larger, the optical resolution of the decoder output will deteriorate because of the reduced NA of the communication system.”

7. For the encoder in this manuscript, we already know the diffractive media, and if we also know the size, shape, and positions of the occlusions, we can directly calculate the phase pattern to set on the encoder. This is similar to many other works.

-- We kindly disagree here and thank the referee for giving us the opportunity to clarify a few important points:

1- We would like to emphasize that the diffractive decoder is not known a priori before the design

process. The diffractive decoder complements the electronic encoder as part of the encoding/decoding scheme and is trained jointly with the encoder neural network for communicating around opaque occlusions. In other words, the diffractive decoder, together with the encoder, evolves during the training process.

2- Our framework is robust against random and unknown changes in the shape/size of the occlusion and the 3D positions of the diffractive decoder layers and, therefore, cannot be regarded as a simple phase encoder, as the referee's comments above claim.

To better clarify these points for our readers, we have performed additional experiments to showcase the robustness of our approach to unknown and random changes in the opaque occlusion. **New Supplementary Videos 1-3 and Fig. 10** have been added to showcase these results and analyses, together with a new subsection 'Dynamic occlusions' in the 'Results' section of our revised manuscript, quoted below:

*"...So far, we have analyzed our framework for static occlusions that do not change over time. Here, we demonstrate the adaptability of our framework to situations where the occlusion shape/size can randomly change over time without our knowledge. In other words, we design encoder-decoder pairs which can communicate around opaque occlusions of varying unknown shapes, without any change to the encoder or the decoder. In **Supplementary Video 1**, we present our analysis depicting the performance of an $L = 3$ design as the shape of the opaque occlusion randomly changes within the occlusion plane. For training and testing of this design, the occlusion shape was parameterized by $r_{max} \approx 17.6\lambda$, where the radii of the partial circles comprising the occlusions were randomly drawn from the distribution $Uniform(9r_{max}/11, r_{max})$; see the Methods section for details. As shown in **Supplementary Video 1**, the same electronic encoder and diffractive decoder successfully communicate the desired images of the objects even if the occlusion changes randomly. In **Supplementary Videos 2 and 3**, we show two additional $L = 3$ designs with $r_{max} \approx 29.3\lambda$ and $r_{max} \approx 41.1\lambda$, respectively, showcasing a decent performance despite a significant loss of information caused by the different opaque occlusions of varying random and unknown shapes.*

*We also experimentally demonstrated robust communication around opaque occlusions of varying, random shapes ($r_{max} \approx 18.1\lambda$) at $\lambda = 0.75\text{mm}$, using a fixed encoder-decoder pair with $L = 1$. The results of these experiments are presented in **Fig. 10**, where all the desired objects of interest were successfully communicated around two different occlusions by the same encoder-decoder design comprising a single-diffractive layer..."*

To further address the reviewer's concern, we have performed additional analyses on the effect of lateral and axial random displacements and uncertainties of the occlusion as well as the diffractive layers. We have also outlined strategies to make the design robust to such physical random misalignments and uncertainties. We have **included four new supplementary figures, Supplementary Figs. 3-6**, and added a new subsection 'Impact of misalignments on performance' in the 'Results' section of our revised manuscript, quoted below:

"...Next, we focus on the effect of physical misalignments on the performance of our framework for communication around opaque occlusions. First, we explore the effect of random misalignments of the physical layers of the diffractive decoder. For this analysis, we model the misalignments of the layers using random variables $\delta_{x,l} \sim Uniform(-\delta_{lat}, \delta_{lat})$, $\delta_{y,l} \sim Uniform(-\delta_{lat}, \delta_{lat})$ and

$\delta_{z,l} \sim \text{Uniform}(-\delta_{ax}, \delta_{ax})$ where $\delta_{x,l}$, $\delta_{y,l}$ and $\delta_{z,l}$ denote the displacements of the diffractive layer l from its nominal position along x , y and z directions, respectively; $l = 1, \dots, L$. δ_{lat} and δ_{ax} are the parameters quantifying the degree of the lateral and axial random misalignments, respectively. In **Supplementary Fig. S3**, we present the effect of only the lateral random misalignments of the diffractive layers ($\delta_{lat} \geq 0$) assuming no axial misalignment ($\delta_{ax} = 0$). These results reveal that the design trained without taking such random lateral misalignments of the layers into consideration ($\delta_{lat,tr} = 0$) fails to successfully communicate through an opaque occlusion when tested with various levels of random lateral misalignments, i.e., $\delta_{lat,te} > 0$. This sensitivity to random physical misalignments can be improved by taking such misalignments into account during the design phase by training with $\delta_{lat,tr} > 0$. We can see from the same **Supplementary Fig. S3** that the performance of the $\delta_{lat,tr} = 4\lambda$ design remains decent up to $\delta_{lat,te} = 8\lambda$, whereas for the $\delta_{lat,tr} = 8\lambda$ design, there is no perceptible degradation in the performance as $\delta_{lat,te}$ goes from 0 to 8λ . **Supplementary Fig. S3b** further reports, as a function of $\delta_{lat,te}$, the average PSNR and average SSIM values for these designs trained with different $\delta_{lat,tr}$, **showing that the resilience of encoder-decoder designs against random lateral misalignments can be significantly improved by training with suitably chosen $\delta_{lat,tr}$, with a modest trade-off in communication performance.**

The same conclusion also holds for axial random misalignments, as shown in Supplementary Fig. S4. It is to be noted that as the resilience to large random axial misalignments (e.g., $\delta_{ax,te} = 8\lambda$) is attained by training with $\delta_{ax,tr} > 0$, the decrease in performance for no misalignments ($\delta_{ax,te} = 0$) is virtually negligible, which is highly desired.

Following a similar strategy, our jointly trained encoder-decoder pair designs can also be made resilient to lateral and axial random displacements of the opaque occlusion as illustrated in Supplementary Figs. S5 and S6...

8. The effectiveness of the machine learning (ML) approach employed here is also a question. If we want to use generated data for training, the requirement of matching between the real world and the virtual model needs to meet high levels of accuracy, which might only be achievable in the laboratory, as demonstrated in the manuscript. Alternatively, direct feedback communication channels need to be established from the output to the encoder. However, given the amount of training/testing data required for each configuration and the dynamics of real-world experiments, this approach could undermine the purpose of free-space communication around arbitrary opaque occlusions.

-- We thank the reviewer for these thoughtful comments. All of these concerns have been addressed in our revised manuscript.

To address the reviewer's concern, we have performed additional analyses on the effect of lateral and axial random displacements of the occlusion as well as the diffractive layers. We have also outlined strategies to make the design robust to such physical random misalignments. We have **included four new supplementary figures, Supplementary Figs. 3-6**, and added a new subsection 'Impact of misalignments on performance' in the 'Results' section of our revised manuscript, quoted below:

"...Next, we focus on the effect of physical misalignments on the performance of our framework for communication around opaque occlusions. First, we explore the effect of random misalignments of

the physical layers of the diffractive decoder. For this analysis, we model the misalignments of the layers using random variables $\delta_{x,l} \sim \text{Uniform}(-\delta_{lat}, \delta_{lat})$, $\delta_{y,l} \sim \text{Uniform}(-\delta_{lat}, \delta_{lat})$ and $\delta_{z,l} \sim \text{Uniform}(-\delta_{ax}, \delta_{ax})$ where $\delta_{x,l}$, $\delta_{y,l}$ and $\delta_{z,l}$ denote the displacements of the diffractive layer l from its nominal position along x , y and z directions, respectively; $l = 1, \dots, L$. δ_{lat} and δ_{ax} are the parameters quantifying the degree of the lateral and axial random misalignments, respectively. In **Supplementary Fig. S3**, we present the effect of only the lateral random misalignments of the diffractive layers ($\delta_{lat} \geq 0$) assuming no axial misalignment ($\delta_{ax} = 0$). These results reveal that the design trained without taking such random lateral misalignments of the layers into consideration ($\delta_{lat,tr} = 0$) fails to successfully communicate through an opaque occlusion when tested with various levels of random lateral misalignments, i.e., $\delta_{lat,te} > 0$. This sensitivity to random physical misalignments can be improved by taking such misalignments into account during the design phase by training with $\delta_{lat,tr} > 0$. We can see from the same **Supplementary Fig. S3** that the performance of the $\delta_{lat,tr} = 4\lambda$ design remains decent up to $\delta_{lat,te} = 8\lambda$, whereas for the $\delta_{lat,tr} = 8\lambda$ design, there is no perceptible degradation in the performance as $\delta_{lat,te}$ goes from 0 to 8λ . **Supplementary Fig. S3b** further reports, as a function of $\delta_{lat,te}$, the average PSNR and average SSIM values for these designs trained with different $\delta_{lat,tr}$, **showing that the resilience of encoder-decoder designs against random lateral misalignments can be significantly improved by training with suitably chosen $\delta_{lat,tr}$, with a modest trade-off in communication performance.**

The same conclusion also holds for axial random misalignments, as shown in Supplementary Fig. S4. It is to be noted that as the resilience to large random axial misalignments (e.g., $\delta_{ax,te} = 8\lambda$) is attained by training with $\delta_{ax,tr} > 0$, the decrease in performance for no misalignments ($\delta_{ax,te} = 0$) is virtually negligible, which is highly desired.

Following a similar strategy, our jointly trained encoder-decoder pair designs can also be made resilient to lateral and axial random displacements of the opaque occlusion as illustrated in Supplementary Figs. S5 and S6...

Moreover, we have performed additional experiments to showcase the robustness of our approach to unknown and random changes in the opaque occlusion. **New Supplementary Videos 1-3 and Fig. 10** have been added to showcase these results and analyses, together with a new subsection 'Dynamic occlusions' in the 'Results' section of our revised manuscript, quoted below:

"...So far, we have analyzed our framework for static occlusions that do not change over time. Here, we demonstrate the adaptability of our framework to situations where the occlusion shape/size can randomly change over time without our knowledge. In other words, we design encoder-decoder pairs which can communicate around opaque occlusions of varying unknown shapes, without any change to the encoder or the decoder. In **Supplementary Video 1**, we present our analysis depicting the performance of an $L = 3$ design as the shape of the opaque occlusion randomly changes within the occlusion plane. For training and testing of this design, the occlusion shape was parameterized by $r_{max} \approx 17.6\lambda$, where the radii of the partial circles comprising the occlusions were randomly drawn from the distribution $\text{Uniform}(9r_{max}/11, r_{max})$; see the Methods section for details. As shown in **Supplementary Video 1**, the same electronic encoder and diffractive decoder successfully communicate the desired images of the objects even if the occlusion changes randomly. In **Supplementary Videos 2 and 3**, we show two additional $L = 3$ designs with $r_{max} \approx 29.3\lambda$ and

$r_{max} \approx 41.1\lambda$, respectively, showcasing a decent performance despite a significant loss of information caused by the different opaque occlusions of varying random and unknown shapes.

We also experimentally demonstrated robust communication around opaque occlusions of varying, random shapes ($r_{max} \approx 18.1\lambda$) at $\lambda = 0.75\text{mm}$, using a fixed encoder-decoder pair with $L = 1$. The results of these experiments are presented in **Fig. 10**, where all the desired objects of interest were successfully communicated around two different occlusions by the same encoder-decoder design comprising a single-diffractive layer..."

These results and analyses ascertain that our framework could handle the dynamics of real-world without the need for establishing a feedback communication channel from the output to the encoder.

9. The experiment can be easily conducted using visible light, a phase-only spatial light modulator (SLM), a meta-surface (diffractive encoder), and a camera. Then possibly shows much more significant results compared to scattering media or diffusive reflection surface or lens, if any.

-- Due to the significance and unique properties of THz waves for next-generation communication and free-space systems, we have selected THz as the spectral band for the experimental proof of concept demonstrations reported in our results. Our results and analyses follow Maxwell's equations, and therefore our framework is **universally** applicable to any part of the electromagnetic spectrum.

The arguments/discussions around which part of the spectrum is the most valuable, significant one and for which application are beyond the scope of this manuscript. For example, if you ask the defense industry vs. the entertainment industry, one could very well argue different spectral bands being more valuable. Therefore, the demonstrations of our approach at other spectral bands of interest are outside the scope of this manuscript and left as future work.

REVIEWER COMMENTS

Reviewer #1 (Remarks to the Author):

I am pleased with the authors' replies. I think it is an interesting idea with solid demonstration. I support the publication of the work.

Reviewer #2 (Remarks to the Author):

The revision addresses and resolves the technical questions I had regarding the initial version, in particular the occluder size issue and the dependency on occluder position.

That said, I still remain unconvinced about the impact of the work -- as pointed out in the initial review, the setting of projecting spatially resolved information over a distance for communications purposes seems highly contrived, and I don't see a scenario where this is preferable over temporal encodings. The tolerable occluder size is so small in absolute terms that the system robustness would suffer outside a controlled lab setting (in which case, why would there be an occluder to begin with).

Reviewer #3 (Remarks to the Author):

I genuinely appreciate the authors taking the criticisms seriously and doing the work to address them.

I would honestly still like a bit more discussion about the differences and similarities between measuring the transmission matrix and using a neural network, but I don't think it is worth another round of revision.

I recommend publication.

Reviewer #4 (Remarks to the Author):

The revised manuscript is much clearer than the first version. Thank the authors for making this. I still feel not yet convinced by the demonstration and explanation.

1. Agree that the diffraction at the edge of the occlusion does contribute to the signal at the output plane. The main contribution of the signal should still be the direct communication between the encoder and decoder, but the work presents as magic to go around the occlusion. Several demonstrations imply this such as:

a. The demonstration with the static occlusion of $1.15 w_c$, the performance degrades significantly.

b. The demonstration with dynamically changing occlusions, the occlusion size is significantly smaller than w_c ($r_{max} = 44 \lambda$ and $w_c = 64 \lambda$), therefore, a very large amount of communication is through the direct channel from the encoder to decoder. If the demonstration with occlusion of $1.15 w_c$, it is hard to succeed.

c. Same thing for demonstration with displacement, the occlusion size is only 50% of w_c (the blocking area is only 25%), which means a significant amount (75%) of information can go directly.

2. With the point above, the work becomes pretty straightforward, as per my previous comments. The binary image with the sparsity of the information source makes things easier.
3. In fact, with the optical model, one can easily build the transfer matrix from encoder to decoder and do the job with direct calculations. In fact, the ML algorithm is a way of learning/measuring the transfer matrix. So, the novelty of the work is the design of the decoder hardware to make the work done more efficiently as demonstrated by authors.
4. Again, I also feel that the communication context in this manuscript is too impractical. Not sure in which circumstance, one would use this.

We sincerely thank the referees for their reviews and the constructive feedback that we have received on our manuscript “**Learning Diffractive Optical Communication Around Arbitrary Opaque Occlusions**” submitted to *Nature Communications*.

As detailed below, we have revised our manuscript in response to the reviewers’ comments. The original referee comments are shown in black color, whereas for ease of communication, our answers are provided in blue. The changes that we have made in the manuscript text are highlighted in yellow.

Summary of our Revisions:

In response to the reviewers’ comments, we have added new discussions highlighting additional applications of our framework and clarifying the differences between our method and transmission matrix approach. These changes have been marked in yellow in the Discussion section of the revised manuscript.

Reviewer #1 (Remarks to the Author):

I am pleased with the authors' replies. I think it is an interesting idea with solid demonstration. I support the publication of the work.

-- We sincerely thank the reviewer for the positive and constructive evaluations.

Reviewer #2 (Remarks to the Author):

The revision addresses and resolves the technical questions I had regarding the initial version, in particular the occluder size issue and the dependency on occluder position.

That said, I still remain unconvinced about the impact of the work -- as pointed out in the initial review, the setting of projecting spatially resolved information over a distance for communications purposes seems highly contrived, and I don't see a scenario where this is preferable over temporal encodings. The tolerable occluder size is so small in absolute terms that the system robustness would suffer outside a controlled lab setting (in which case, why would there be an occluder to begin with).

-- We sincerely thank the reviewer for the constructive feedback in both rounds of the peer review process. To address the reviewer’s concern, we have added the following text to the **Discussion** section to better highlight potential additional applications and the impact of our framework:

“...Therefore, one of the important contributions of this work has been to establish an electronic-optical encoder-decoder communication pair that uses a string of scattering edges resulting from the topology of an opaque occlusion, forming a 3D loop of secondary waves. Several examples that we considered in this work solely used these strings of secondary waves in the form of edge scattering functions as the main source of optical information transmission; see, for example, Figs. 2a, 4 and 5 where $w_o > w_c$ and the opaque occlusion body entirely blocks the direct line-of-sight between the transmitter and the receiver apertures. Our framework

can successfully transfer the target spatial information even in these cases, where the only communication channel between the transmitter and the receiver is the scattering from the occlusion edges.

Beyond optical image transmission around opaque occlusions, various other applications can potentially be enabled by the presented framework operating at different parts of the electromagnetic spectrum. **For example, several mobile units/agents (such as autonomous robots) within a certain output region can be dynamically targeted even in the presence of occlusions that block the direct line-of-sight between the encoder/transmitter and the mobile receivers.** In this scenario, the encoder can be dynamically updated to deliver optical radiation/power or information of interest to these mobile units/agents that are free to move within an output FOV. **Another application of the presented concept could arise in the operation of wearable and implantable devices, which need to blend miniaturized optics and electronics for their operation.** In such a scenario, our scheme could be useful, for example, to optically power/excite an array of implanted optical sensors by passing light around occlusions arising from the metal electronics or other opaque parts of the wearable/implantable system. **Even more challenging applications could be envisioned using the presented framework to enable, for example, the visualization and detection of hidden objects sandwiched between two opaque occlusions, such as objects located between two walls or metal screens.** In such applications, the first edge scattering function of the first opaque occlusion can be used by the encoder network to illuminate the hidden objects behind the first occlusion, whereas the second edge scattering function of the second opaque occlusion can be used to transmit the hidden objects' optical information to a jointly trained diffractive decoder network for all-optically revealing/reconstructing the information of the objects sandwiched between the opaque screens. **This could have major implications for e.g., security and defense applications, enabling us to see/detect objects hidden between metal plates or partial walls.** In such applications where the string of edge scattering function of each opaque occlusion is cascaded with the other edge scattering functions of successive opaque occlusions, the detection signal-to-noise ratio sets practical challenges, demanding high-power encoders/transmitters and high sensitivity output detectors.”

Reviewer #3 (Remarks to the Author):

I genuinely appreciate the authors taking the criticisms seriously and doing the work to address them. I would honestly still like a bit more discussion about the differences and similarities between measuring the transmission matrix and using a neural network, but I don't think it is worth another round of revision.

I recommend publication.

-- We wholeheartedly thank the reviewer for the positive evaluation and recommendation. We also took this 2nd revision opportunity to better address the referee's suggestion. Accordingly, we have added the following Discussion paragraph to clarify the differences between the transmission matrix-based approaches and our work:

“... We would also like to highlight some key differences between our approach and the transmission matrix-based approaches used to control light propagation through scattering media⁴⁴. A transmission matrix-based approach relates the optical field at the receiver to the field at the transmitter in the presence of scattering, which can be measured or approximated. However, without the use of an optimized diffractive optical decoder

architecture, the sole knowledge or accurate approximation of such a transfer matrix does not lend itself to the successful transfer of images or spatial information of interest around opaque occlusions with zero light transmittance; for example, Supplementary Fig. S11 illustrates that without an optimized diffractive decoder at the back-end, just an encoder optimization even with the precise knowledge of the transfer matrix of the system cannot perform successful image transfer around opaque occlusions. Moreover, our joint training approach for optimizing the electronic encoder – diffractive decoder pair is accomplished using the angular spectrum approach, which seamlessly blends all the propagating modes of optical information into our training. Using a wave propagation model instead of a known transmission matrix allows us to statistically incorporate various deviations in the forward model, which might randomly occur in real-world situations; this “vaccination” based training strategy builds design resilience against such random deviations in the physical system, as shown in Supplementary Figs. S3-S6 and Supplementary Videos 1-3. Measuring the transmission matrices corresponding to all the forward model states resulting from these random deviations would be impractical and, more importantly, would still not reveal competitive image transmission behavior around opaque occlusions without the use of a jointly trained diffractive optical decoder (see Supplementary Fig. S11). It should also be noted that the encoder neural network within our framework is not a surrogate for the transmission matrix. In fact, given the knowledge of how the optical field is scattered by the edges of an opaque occlusion located between the transmitter and the receiver, the electronic encoder learns an object representation model that successfully uses the edge scattering function of the occlusion to deliver the optical information to a jointly-trained diffractive decoder that all-optically converts this encoded and scattered information back to the desired object representation at the output, bypassing the zero transmission occlusion body.”

Reviewer #4 (Remarks to the Author):

The revised manuscript is much clearer than the first version. Thank the authors for making this.

-- We sincerely thank the reviewer for the positive evaluation.

I still feel not yet convinced by the demonstration and explanation.

1. Agree that the diffraction at the edge of the occlusion does contribute to the signal at the output plain. The main contribution of the signal should still be the direct communication between the encoder and decoder, but the work presents as magic to go around the occlusion. Several demonstrations imply this such as:

- a. The demonstration with the static occlusion of $1.15w_c$, the performance degrades significantly.
- b. The demonstration with dynamically changing occlusions, the occlusion size is significantly smaller than w_c ($r_{max} = 44\lambda$ and $w_c = 64\lambda$), therefore, a very large amount of communication is through the direct channel from the encoder to decoder. If the demonstration with occlusion of $1.15w_c$, it is hard to succeed.
- c. Same thing for demonstration with displacement, the occlusion size is only 50% of w_c (the blocking area is only 25%), which means a significant amount (75%) of information can go directly.

-- We thank the reviewer for this constructive and valuable feedback.

Regarding the case where the static occlusion has $w_o \approx 1.15w_c$, we kindly refer the referee to **Figs. 2 and 4**; the performance/quality of our image reconstructions under $w_o \approx 1.15w_c$ is indeed outstanding

for all the 24 image examples shown, which contradict the referee's comments.

To better emphasize these points and clarify some of the messages in our manuscript, we have added the following paragraph in our **Discussion** section:

“...Therefore, one of the important contributions of this work has been to establish an electronic-optical encoder-decoder communication pair that uses a string of scattering edges resulting from the topology of an opaque occlusion, forming a 3D loop of secondary waves. Several examples that we considered in this work solely used these strings of secondary waves in the form of edge scattering functions as the main source of optical information transmission; see, for example, Figs. 2a, 4 and 5 where $w_o > w_c$ and the opaque occlusion body entirely blocks the direct line-of-sight between the transmitter and the receiver apertures. Our framework can successfully transfer the target spatial information even in these cases, where the only communication channel between the transmitter and the receiver is the scattering from the occlusion edges.

In short, our work demonstrated several challenging cases where significant information loss occurs by extended zero-transmission opaque occlusions with $w_o > w_c$, the complexity of which is evident from the failure of other alternative approaches, as shown in **Supplementary Fig. 11**.

2. With the point above, the work becomes pretty straightforward, as per my previous comments. The binary image with the sparsity of the information source makes things easier.

-- Contrary to these comments, **we clearly showed outstanding image reconstruction results corresponding to grayscale and non-sparse images for $w_o > w_c$ in Figure 4**. We also kindly disagree with the assertion about the task being straightforward. We believe that the failure of several alternative schemes, as presented in **Supplementary Fig. 11**, clearly proves the complexity of our tasks.

3. In fact, with the optical model, one can easily build the transfer matrix from encoder to decoder and do the job with direct calculations. In fact, the ML algorithm is a way of learning/measuring the transfer matrix. So, the novelty of the work is the design of the decoder hardware to make the work done more efficiently as demonstrated by authors.

-- To better clarify the unique contributions of this work, we have added new text to our **Discussion** paragraph, quoted below. These points clearly address the referee's comments:

“...We would also like to highlight some key differences between our approach and the transmission matrix-based approaches used to control light propagation through scattering media⁴⁴. A transmission matrix-based approach relates the optical field at the receiver to the field at the transmitter in the presence of scattering, which can be measured or approximated. **However, without the use of an optimized diffractive optical decoder architecture, the sole knowledge or accurate approximation of such a transfer matrix does not lend itself to the successful transfer of images or spatial information of interest around opaque occlusions with zero light transmittance; for example, Supplementary Fig. S11 illustrates that without an optimized diffractive decoder at the back-end, just an encoder optimization even with the precise knowledge of the transfer matrix of the system cannot perform successful image transfer around opaque occlusions.** Moreover, our joint training approach for optimizing the electronic encoder – diffractive decoder pair is accomplished using

the angular spectrum approach, which seamlessly blends all the propagating modes of optical information into our training. **Using a wave propagation model instead of a known transmission matrix allows us to statistically incorporate various deviations in the forward model, which might randomly occur in real-world situations; this “vaccination” based training strategy builds design resilience against such random deviations in the physical system, as shown in Supplementary Figs. S3-S6 and Supplementary Videos 1-3. Measuring the transmission matrices corresponding to all the forward model states resulting from these random deviations would be impractical and, more importantly, would still not reveal competitive image transmission behavior around opaque occlusions without the use of a jointly trained diffractive optical decoder (see Supplementary Fig. S11).** It should also be noted that the encoder neural network within our framework is not a surrogate for the transmission matrix. In fact, given the knowledge of how the optical field is scattered by the edges of an opaque occlusion located between the transmitter and the receiver, the electronic encoder learns an object representation model that successfully uses the edge scattering function of the occlusion to deliver the optical information to a jointly-trained diffractive decoder that all-optically converts this encoded and scattered information back to the desired object representation at the output, bypassing the zero transmission occlusion body.”

4. Again, I also feel that the communication context in this manuscript is too impractical. Not sure in which circumstance, one would use this.

-- To address the reviewer's comments, we have added the following text to the **Discussion** section to highlight potential additional applications and the impact of our framework:

“...Therefore, one of the important contributions of this work has been to establish an electronic-optical encoder-decoder communication pair that uses a string of scattering edges resulting from the topology of an opaque occlusion, forming a 3D loop of secondary waves. Several examples that we considered in this work solely used these strings of secondary waves in the form of edge scattering functions as the main source of optical information transmission; see, for example, Figs. 2a, 4 and 5 where $w_o > w_c$ and the opaque occlusion body entirely blocks the direct line-of-sight between the transmitter and the receiver apertures. Our framework can successfully transfer the target spatial information even in these cases, where the only communication channel between the transmitter and the receiver is the scattering from the occlusion edges.

Beyond optical image transmission around opaque occlusions, various other applications can potentially be enabled by the presented framework operating at different parts of the electromagnetic spectrum. **For example, several mobile units/agents (such as autonomous robots) within a certain output region can be dynamically targeted even in the presence of occlusions that block the direct line-of-sight between the encoder/transmitter and the mobile receivers.** In this scenario, the encoder can be dynamically updated to deliver optical radiation/power or information of interest to these mobile units/agents that are free to move within an output FOV. **Another application of the presented concept could arise in the operation of wearable and implantable devices, which need to blend miniaturized optics and electronics for their operation.** In such a scenario, our scheme could be useful, for example, to optically power/excite an array of implanted optical sensors by passing light around occlusions arising from the metal electronics or other opaque parts of the wearable/implantable system. **Even more challenging applications could be envisioned using the presented framework to enable, for example, the visualization and detection of hidden objects sandwiched between two opaque occlusions, such as objects located between two walls or metal screens.** In such applications, the first edge scattering function of the first opaque occlusion can be used by the encoder

network to illuminate the hidden objects behind the first occlusion, whereas the second edge scattering function of the second opaque occlusion can be used to transmit the hidden objects' optical information to a jointly trained diffractive decoder network for all-optically revealing/reconstructing the information of the objects sandwiched between the opaque screens. **This could have major implications for e.g., security and defense applications, enabling us to see/detect objects hidden between metal plates or partial walls.** In such applications where the string of edge scattering function of each opaque occlusion is cascaded with the other edge scattering functions of successive opaque occlusions, the detection signal-to-noise ratio sets practical challenges, demanding high-power encoders/transmitters and high sensitivity output detectors.”

REVIEWERS' COMMENTS

Reviewer #4 (Remarks to the Author):

Thank the authors for their efforts in revising the manuscript. I am convinced of the technique demonstration with small occlusion, both static and dynamic occlusion, which is understandable due to direct communication. Other than this, it is not convincing.

For large occlusion, the author responded with theoretical demonstrations (Fig 2 and 4), while the practical demonstration was not for this case. As mentioned last round, it is challenging and likely that it is impractical. In fact, for demonstration in Fig. 9, the authors should present the experiment configuration where the occlusion width is presented in terms of W_c . Such important information is well stated in the simulation (0.5, 0.8, 1.15 W_c) but, somehow, hidden in the experiment. From the picture, I guess the occlusion width is only about 50% of the W_c , i.e., 25% of the critical area. Therefore, there is still 75% of the area for direct transmission, which is huge. I expect the authors to clearly state these numbers and double-check my estimation.

For the dynamic occlusion, both simulation and demonstration are with small occlusion (seems similar to static occlusion in the experiment. As stated above, the relative area for direct communication is 75%, while the blocking area is 25%. It implies that the number of direct channels is 3 times greater than that of blocking channels. It is very hard to imagine that it would work if the occlusion completely blocked the direct channels.

We sincerely thank the reviewer for the constructive feedback that we have received on our manuscript “**Learning Diffractive Optical Communication Around Arbitrary Opaque Occlusions**” submitted to *Nature Communications*.

Our responses to the reviewer's remarks are provided below. The reviewer comments are shown in black color, whereas for ease of communication, our responses are shown in blue.

Reviewer #4:

Thank the authors for their efforts in revising the manuscript. I am convinced of the technique demonstration with small occlusion, both static and dynamic occlusion, which is understandable due to direct communication.

-- We thank the reviewer for his/her valuable time and constructive and positive feedback.

Other than this, it is not convincing. For large occlusion, the author responded with theoretical demonstrations (Fig 2 and 4), while the practical demonstration was not for this case. As mentioned last round, it is challenging and likely that it is impractical. In fact, for demonstration in Fig. 9, the authors should present the experiment configuration where the occlusion width is presented in terms of W_c . Such important information is well stated in the simulation (0.5, 0.8, 1.15 W_c) but, somehow, hidden in the experiment. From the picture, I guess the occlusion width is only about 50% of the W_c , i.e., 25% of the critical area. Therefore, there is still 75% of the area for direct transmission, which is huge. I expect the authors to clearly state these numbers and double-check my estimation.

-- We thank the reviewer for giving us the opportunity to clarify.

Regarding Fig. 9, we would like to highlight that the information that the reviewer presumed to be “somehow hidden” is actually mentioned in the first sentence of the same figure caption: “Experimental results with an $L = 1$ design for an occlusion width of $w_o = 32\lambda \approx 0.5w_c$ operating at a wavelength of $\lambda = 0.75\text{mm}$.” We believe this statement is clear enough for the readers.

We would also like to assert that, with sufficient detection SNR, the experimental success for $w_o = 32\lambda \approx 0.5w_c$ should translate to larger occlusions, as clearly evidenced by our numerical results presented in the main text: **Figs. 2, 4 and 5**.

For the dynamic occlusion, both simulation and demonstration are with small occlusion (seems similar to static occlusion in the experiment. As stated above, the relative area for direct communication is 75%, while the blocking area is 25%. It implies that the number of direct channels is 3 times greater than that of blocking channels. It is very hard to imagine that it would work if the occlusion completely blocked the direct channels.

-- We kindly disagree with the reviewer since his/her claims about the occlusion size for the simulation of dynamic occlusions are clearly in contradiction with the analyses that we provided in our **Supplementary Videos 2 and 3**.

To facilitate our response, we first refer to the ‘Dynamic occlusion modeling’ subsection of the ‘Methods’ section of our manuscript:

“... *The dynamic occlusions were modeled as the union of 24 concentric and disjoint partial circles extending equal angles ($360^\circ/24 = 15^\circ$) at the center. The radii of these partial circles were randomly sampled independently from the distribution $Uniform(9r_{max}/11, r_{max})$, where the parameter r_{max} is a*

measure of the level of the opaque occlusion. At each training iteration, the occlusion was randomly sampled from the distribution defined by r_{max} .”

Accordingly, the maximum occlusion area can be πr_{max}^2 . In the following table, we list the maximum occlusion area as a fraction of the critical area w_c^2 , where $w_c \approx 64.95\lambda$ for the dynamic occlusion results presented in **Supplementary Videos 1-3**:

	r_{max}	$\frac{\pi r_{max}^2}{w_c^2}$
Supplementary Video 1	17.6λ	0.23
Supplementary Video 2	29.3λ	0.64
Supplementary Video 3	41.1λ	1.26

Clearly, the $\frac{\pi r_{max}^2}{w_c^2}$ values of 0.64 and 1.26 corresponding to Supplementary Videos 2 and 3, respectively, refute the reviewer’s statement: “...*For the dynamic occlusion, both simulation and demonstration are with small occlusion*”.